# Biomimetic Bouligand chiral fibers array enables strong and superelastic ceramic aerogels

Hongxing Wang[1], Longdi Cheng[1], Jianyong Yu [1,2], Yang Si [1,2] ✉ &
Bin Ding [1,2] ✉

Ceramic aerogels are often used when thermal insulation materials are desired; however, they are still plagued by poor mechanical stability under thermal shock. Here, inspired by the dactyl clubs of mantis shrimp found in nature, which form by directed assembly into hierarchical, chiral and Bouligand (twisted plywood) structure exhibiting superior mechanical properties, we present a compositional and structural engineering strategy to develop strong, superelastic and fatigue resistance ceramic aerogels with chiral fibers array resembling Bouligand architecture. Benefiting from the stress dissipation, crack torsion and mechanical reinforcement of micro-/nano-scale Bouligand array, the tensile strength of these aerogels (170.38 MPa) is between one and two orders of magnitude greater than that of state-of-the-art nanofibrous aerogels. In addition, the developed aerogels feature low density and thermal conductivity, good compressive properties with rapid recovery from 80 % strain, and thermal stability up to 1200 °C, making them ideal for thermal insulation applications.

Ceramic aerogels exhibit a combination of low density (0.003–0.5 g cm$^{-3}$), low thermal conductivity (0.012 W m$^{-1}$ K$^{-1}$ to 0.033 W m$^{-1}$ K$^{-1}$ in air), good stability in extreme environments (−196 °C to 1200 °C) and a range of functional capabilities, such as the ability to restrict the flow of liquids or gases, which make them ideal materials for a variety of applications, including thermal and electrical engineering, catalysis, optics, and aerospace industry[1–3]. However, with the ongoing exploration of deep Earth (Antarctica) and deep space (the Moon and Mars), the large thermal gradient is a fundamental barrier to further discovery. In particular, traditional ceramic aerogel materials featuring inherent "pearl necklace" architecture, a network of ceramic nanoparticles connected by thin particle–particle necks, are prone to brittleness and structural instability. Hence, these conditions pose additional challenges for ceramic aerogels[4,5]. Tremendous improvements have been developed to enhance the poor mechanical properties, focusing particularly on the introduction of continuous structural units (nanofibers or nanosheets) as building blocks and designing

them with specific microstructures[6–8], e.g., SiO$_2$ nanofibrous aerogels[1], ZrO$_2$–Al$_2$O$_3$ nanofibrous aerogels[2], mullite-based nanofibrous aerogels[9], SiC@SiO$_2$ nanowire aerogels[10], and so forth. These continuous units, combined with the microstructures that eliminate traditional necklace-like structures and other faults, offer extra degrees of freedom to decrease the risk of brittle failure in interconnected building blocks, resulting in some improvement in compressive properties[6–10]. Further advances used reaction electrospinning to construct three-dimensional (3D) woven crimped nanofiber architectures for improved mullite nanofiber aerogels with a tensile strain of 100% and a tensile strength of 12.7 kPa[8]. The tensile properties were mainly derived from the crimped, strong and flexible nanofibers, effectively preventing the premature breakage of nanofibers and the stress diffusion effect of crimped structure[7,8]. Nevertheless, these methods predominantly involved minor modifications of building blocks on a limited scale (nanoscale) and rather simplistic architectures. As a result, at present, all ceramic fibrous aerogel materials

[1]State Key Laboratory for Modification of Chemical Fibers and Polymer Materials, College of Textiles, Donghua University, 201620 Shanghai, China. [2]Innovation Center for Textile Science and Technology, Donghua University, 200051 Shanghai, China. ✉e-mail: yangsi@dhu.edu.cn; binding@dhu.edu.cn

exhibit low tensile strength (<1 MPa), which limits their effective use under extreme conditions, such as certain applications in the aerospace and defense industries[1–5,8]. Hence, it has been very challenging, if not impossible, to create ceramic aerogels that possess both super-elasticity and high tensile strength[11].

Wood is light and strong; muscles are soft and tough; nacres are hard and resilient[12–14]. The remarkable mechanical properties of natural materials have been the subject of extensive research in the field of materials science[15]. In addition to these well-known biomimetic design paradigms, one particularly impressive example is the Bouligand chiral structure, which contributes to substantial strength in crustaceans (mantis shrimp, lobsters, crabs, beetles, etc.) and is characterized by chiral helix stacks of fibrillar layers rotated by a twist angle[16–18]. Interestingly, the mechanical properties of these natural materials can be finely tuned by modulating the spatial distribution and orientations of the constitutive chitin nanofibrils[16,19,20].

Previous studies have shown that the Bouligand chiral conformation provides superior energy absorption, effective stress transfer, and the ability to inhibit crack propagation by twisting and reorienting ordered nanofibers under external loads, thereby endowing materials with good mechanical properties including toughness, fracture resistance, and impact resistance[19,20]. In addition, the transition from longitudinally staggered array to Bouligand chiral array leads to a monotonic decrease in anisotropic mechanical properties, e.g., the anisotropy ratio of Young's modulus changes from 4.25 to 1.25[16,21]. Other studies have demonstrated that materials with a Bouligand array have band gaps at frequencies associated with the impact stress pulse and therefore confer wave-filtering property[22]. Collectively, these studies support the integration of microfibers to reinforce conventional ceramic nanofibrous aerogels, and form architectures similar to Bouligand chiral arrays that could realize customized strong mechanical performance and other functionalities in ceramic aerogels[23–26].

In this work, we propose a compositional and structural engineering strategy by combining macro-/nano-scale component control over building blocks with precise geometric freedom design in helix chiral fibers array to enable the synchronous fabrication of ceramic fibrous aerogels[13,26]. We assemble biomimetic Bouligand-chiral and fibrous structured ceramic aerogels (BcF-CAs) with ultra-strong, superelastic, and thermal insulation properties[27]. Strikingly, the tensile strength of the obtained biomimetic BcF-CAs is as high as 170.38 MPa, which is one to two orders of magnitude greater than that of state-of-the-art nanofiber aerogel materials[26–28]. In addition, the resulting BcF-CAs exhibit compressive properties of up to 156.47 kPa (at 80% strain), comparable to the performance of most thermal insulation materials. Meanwhile, these aerogels retain 90% of their tensile strength even after calcination at 1200 °C. These results demonstrate that BcF-CAs are optimal candidates for thermal insulation materials for use in demanding applications such as aerospace, power generation, and high-temperature manufacturing processes.

## Results

### Fabrication and Bouligand chiral architecture of BcF-CAs

The mantis shrimp, an appealing but deadly creature. The complex anatomy of the mantis shrimp's claws, consisting of Bouligand chiral stacking of the reinforced chitin protein fibrils, coupled with their exceptional speed, enable them to breach the defenses of even the most resilient prey such as mollusks and crabs[27]. Typically, the unique Bouligand architecture is characterized by a helical array of fibril lamellae with a twisting angle, as shown in Fig. 1a[16–23]. The architecture is subjected to a 180° rotation, which can be quantified by $\alpha$ and $\phi$ along the z-axis direction, where $\alpha$ represents the twisting angle created among two neighboring layers, while $\phi$ defines the twisting angle distribution (Fig. 1b). The direction of the fibers aligned with the x-axis is denoted as $\phi = 0°$, meanwhile the fibers twist counterclockwise around the z-axis[16–18].

This configuration increases the crack surface area and contributes to the reorientation of the fibers in response to external stresses, such as tensile, flexural, and impact loads[29]. Meanwhile, the resulting modulus oscillations within the Bouligand geometry enhance crack torsion (Supplementary Fig. 1)[16,24,29]. Incidentally, the Bouligand chiral array results in in-plane quasi-isotropic mechanical properties (Fig. 1c, d and Supplementary Fig. 2), which overcome the typical limitations of materials with traditional unidirectional 3D fiber structures[16,28–31].

Inspired by the arrangement of fibers in the claws of mantis shrimp, we conceptually designed biomimetic BcF-CAs following the three crucial considerations[26]: (i) the ceramic nanofibers must be flexible, and the reinforced microfibers must be strong[32,33]; (ii) the cross-scale ceramic micro/nanofibers must be assembled into aerogels with Bouligand chiral array; and (iii) the micro/nanofiber skeleton must be bonded and thermally robust[1–3]. The first consideration was fulfilled by adopting a combination of electrospinning and sol–gel methods to produce flexible nanofibers; meanwhile commercial continuous ceramic filaments were used as microfibers for mechanical reinforcement. The second requirement was achieved through a straightforward and reproducible method involving immerse-Bouligand chiral stacking and freeze-shaping. To satisfy the last crucial criterion for stable bonding in fiber skeleton, non-alkaline AlBSi was used as the high-temperature ceramic matrix material[1].

In the proof-of-concept study, mullite nanofibers and $Al_2O_3$ macrofibers were carefully selected as the sample materials due to their thermal stability[1]. Fig. 1e and Supplementary Fig. 4 present the fabrication process of BcF-CAs. The fabrication strategy started with electrospinning mullite/poly (ethylene oxide) sol to produce flexible mullite nanofibers (see Supplementary Methods). The obtained mullite nanofiber had a diameter of 310-420 nm (Supplementary Fig. 5) and a tensile strength of 0.47 MPa (5% strain) (Supplementary Fig. 6). The $Al_2O_3$ microfibers (Supplementary Fig. 7) possessed a fiber diameter of ~7 μm and displayed a tensile strength of 2.1 GPa. Then, the ceramic micro/nanofibers were subjected to immersion in AlBSi sol for 2 h. Notably, AlBSi is known for its thermal stability and mechanical properties, making it a "ceramic glue" for bonding adjacent fibers. This cross-linking method relied on the formation of silicate bonds (X-O-Si), achieved through the calcination of silica nanofibers in the presence of oxygen[1]. Finally, the immersed macrofibers and nanofibers were arranged layer-by-layer with a specific helical angle (0°, 15°, 30°, 75°) and freeze-dried for ~48 h to obtain unbonded BcF-CAs. After annealing in a muffle furnace (900 °C for 1 h in flowing air), bonded BcF-CAs were obtained[1–3,10].

The scanning electron microscopy (SEM) images in Fig. 1f–h clearly depict the unique and intricate biomimetic architecture of the BcF-CAs. The BcF-CAs possessed a tailored Bouligand chiral architecture consisting of both ceramic micro/nanofibers and an interlayer bonding structure. Within this sophisticated architecture, ceramic micro/nanofiber building blocks were packed in a helical arrangement to construct a multilayer structure with the assistance of bonding interlock by AlBSi, which endowed the BcF-CAs with resistance to both mechanical deformation and exposure to extreme temperatures. To investigate the effect of AlBSi, the surface morphology of bondings under a concentration gradient of AlBSi sol (AlBSi- 0.5 wt%, AlBSi-2 wt%, AlBSi-4 wt%, AlBSi-5 wt%, AlBSi-8 wt%) were observed by SEM. As shown in Supplementary Fig. 8, AlBSi agglomerated between the fibers and grew increasingly visible as the concentration increased from 0.5 wt% to 8 wt% AlBSi, and some cracks emerged in samples prepared with 8 wt%. Therefore, an appropriate sol content was mechanically conducive to robust bonding, whereas too high a concentration resulted in an overly rigid adhesive interface, accelerating mechanical failure of BcF-CAs (Supplementary Fig. 9 and Supplementary Fig. 10). Overall, the following investigations were performed with AlBSi sol at a concentration of 5 wt%. Additional morphologies of the

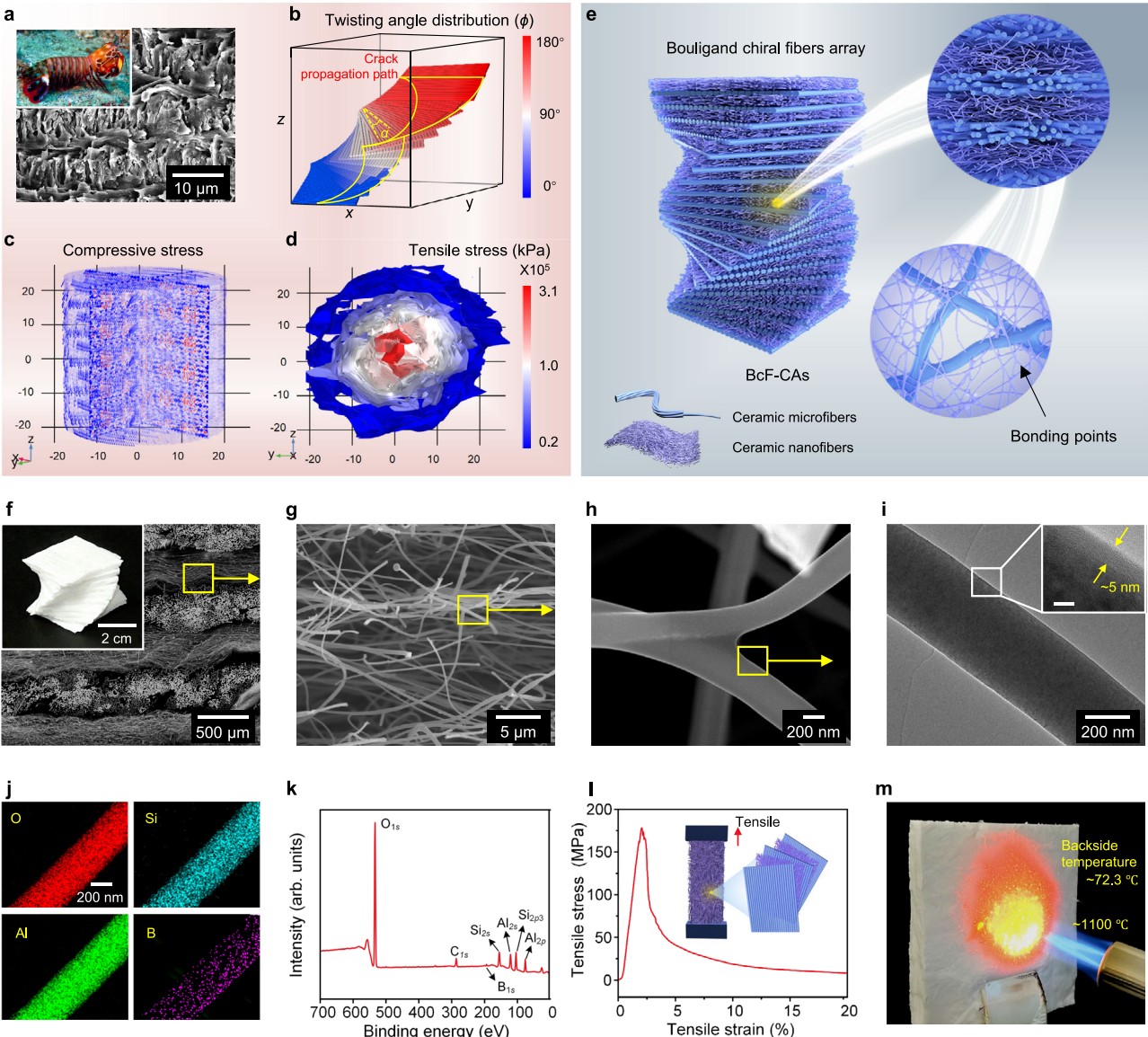

**Fig. 1 | Structural design and fabrication of BcF-CAs. a** Photograph and SEM image of the dactyl club in the mantis shrimp. **b** Structural features and crack propagation path of the typical Bouligand chiral array. Stress distribution of the (**c**) compression and (**d**) tension mode, the color legend of Fig. 1c can be viewed in Supplementary Fig. 3., the color legend of Fig. 1c can be viewed in Supplementary Fig. 3. **e** Schematic illustration of the biomimetic fabrication of the BcF-CAs. **f–h** Photographs and SEM images showing the microscopic architectures of BcF-CAs at different magnifications. **i** A thin coating of AlBSi ceramic on the surface of the fiber, scale bar in inset, 5 nm. **j** HRTEM-EDS images of an individual fiber. **k** XPS spectrum of BcF-CAs for all elements. **l** Tensile stress–strain curve of the BcF-CAs. **m** When a BcF-CA was exposed to a butane blowtorch, no damage was observed.

adhesive region are shown in Supplementary Fig. 11 and Supplementary Fig. 12[1,19].

A pivotal step in this fabrication process was freeze-drying, which eliminated the growth of ice crystals within the pores through air sublimation, thereby preventing capillary forces and allowing for the formation of architecture with ultra-high porousity[1]. Notably, AlBSi was deposited during the freeze-drying process and accumulated on the surfaces of the fibers. Subsequent calcination of the AlBSi resulted in cross-linking, which provided additional structural support that further enhanced the robustness of BcF-CAs. The AlBSi was characterized by a combination of high-resolution transmission electron microscopy (HRTEM) imaging and energy-dispersive X-ray spectroscopy (EDS) mapping. The HRTEM observations revealed a thin bonding layer of ~5 nm (Fig. 1i); meanwhile, EDS mapping results demonstrated that Al and B nearly completely covered O and Si, indicating that the AlBSi constituents were uniformly wrapped and

distributed on the surfaces of the mullite building blocks (Fig. 1j). Indeed, the preparation process for EDS samples involved homogenization and sonication for a prolonged period of 20 min. Despite this rigorous treatment, the AlBSi layer remained attached to the mullite fibers, which suggested that there were stable inter-fiber silicate bonds. Then this strong cross-linked network of BcF-CA was confirmed by X-ray photoelectron spectroscopy (XPS) analysis (Fig. 1k and Supplementary Fig. 13). The characteristic peaks corresponded to tetrahedral conformations of Al- and B-doped silica, and this result was in good agreement with the HRTEM and EDS analysis. Overall, the integrated ceramic feature of the fibers, the AlBSi surface layer and the special complex geometries of the Bouligand array synergistically contributed to the good mechanical properties of the BcF-CAs, which included tensile strength of 170.38 MPa (Fig. 1l)[1]. Finally, the impressive thermal stability of the BcF-CAs was exemplified by their exposure to high-temperature flames (~1100 °C) produced by a butane blow torch.

As expected, the BcF-CAs maintained their shape and structure even under such extreme conditions, which further highlighted their robustness and resilience (Fig. 1m and Supplementary Fig. 14).

### Evaluation of temperature-invariant tensile performance

The stress dissipation behavior associated with the chiral helix structure, coupled with the mechanical reinforcement provided by the microfiber building blocks, contributed to the highly anisotropic mechanical properties of the BcF-CAs, which included ultra-high strength in the vertical helix fiber array direction (i.e., along the fiber direction) and superelasticity along the helix fiber array direction (Supplementary Fig. 15)[30–35]. The tensile properties of the BcF-CAs were investigated in detail. We conducted an evaluation of the two key factors that were responsible for the enhanced tensile mechanical properties of the hierarchical and anisotropic BcF-CAs, the twisting angle ($\alpha$) and the microfiber content ($c$).

First, finite element (FE) simulations using COMSOL Multiphysics were performed to deeply understand the effect of twisting angle on the promotion of the mechanical properties[30]. A series of models that featured different twisting angles were built. As shown in Supplementary Fig. 16a, the schematic diagrams illustrate that the fiber laminas of the chiral array began from $L = 1$ to $L = l + 1$ ($L$ is the number of fiber laminas), the helix of each $L_{th}$ layer formed an angle $\phi = (L-1)\,\alpha$, and $L$ layers were twisted by 180° ($\phi = 180°$ or an integer multiple of 180°) over a pitch distance $D$, where $D = ld$ and $d$ is the thickness of the layer. For example, for $\alpha = 30°$, as shown in Supplementary Fig. 16b, there were six cycles of the same rotation ($l = 6$) of the fiber array around the $z$-axis, indicating that the fiber laminas began from the $x$-axis with distinct rotations for the subsequent six laminas and returned to the $x$-axis for the seventh layer ($L = 7$)[30]. Fig. 2a also illustrates schematic diagrams of the Bouligand array structures. Similarly, for $\alpha = 0°/180°$ (PAR, e. i., it is a parallel fibers array structure), $l = 180$; for $\alpha = 15°$ (BOU-15), $l = 12$; for $\alpha = 30°$ (BOU-30), $l = 6$. Notably, one special model was for $\alpha = 75°$ (BOU-75), that is, a twisting angle of 75° to generate the second lamina and 150° to generate the third lamina until the thirteenth lamina ($L = 13$ and $l = 12$) to make a 900° rotation (five times of 180°)[30].

The simulation results clearly showed that as the value of $\alpha$ decreased from 75° to 0°, the maximum stress (i.e., the stress at initial failure) increased (Fig. 2b). Mechanistically, the Bouligand array enabled the fibers to reorientate in response to external stress. The majority of the fibers reorientate along the tensile direction and underwent tensile deformation owing to stretching/sliding mechanisms, while some of the other fibers rotated symmetrically in the direction away from the stress axis. Remarkably, the smaller $\alpha$ resulted in an increment in the proportion of reinforcing fibers along the direction of tensile loading, thereby enhancing the ductility and toughness of the BcF-CAs to prevent fracture at the macroscopic level (Fig. 2b)[36]. The stress nephograms of the cross sections (PAR, BOU-15, BOU-30, BOU-75) in the parallel fiber layers in Bouligand structures also supported the above conclusions (Fig. 2c and Supplementary Fig. 17a). Additionally, it is noteworthy that for BOU-15, BOU-30 and BOU-75, the maximum stress showed no dependence on the loading direction, demonstrating the in-plane quasi-isotropic tensile mechanical properties of these array models (Fig. 1d). However, the PAR array structure could withstand a high level of loading only when the direction of the force was aligned with the fiber orientation (Supplementary Fig. 17b). That is, the structure was prone to failure when external forces were applied in directions that deviated from the fiber orientation, revealing a major limitation of the PAR arrays. Based upon the above considerations, this study focused on the properties of the Bouligand fiber array with a relative twisting angle $\alpha = 15°$.

To verify the results of the above simulations, the tensile property of BcF-CAs under the same stress for several $\alpha$ values was measured by an Instron 34 TM-5 Universal Machine[37]. Intuitively, the tensile strength increased as $\alpha$ decreased (Fig. 2d top). Furthermore, the BOU-15 subjected to tensile loading in different directions were further investigated (Supplementary Fig. 18). The results indicated that no significant differences in the tensile strength of BcF-CAs were found even when the force direction was varied (Fig. 2d bottom). That is, BcF-CAs with chiral helical fiber arrays had tensile properties that were independent of loading directions and fiber orientation, which was generally difficult to achieve in other fiber-based 3D materials. More importantly, as expected, the experimental behavior correlated well with the simulation results.

In addition, the effect of varying $c$ on the density and tensile strength of the BcF-CAs was measured to choose an optimized $c$ within the Bouligand fiber array (Supplementary Fig. 19)[31]. The results revealed that the BcF-CAs with larger values of $c$ had greater density and strength. Considering the need for ultralight BcF-CAs, the optimum content of microfibers was determined to be 30 vol%. Furthermore, mechanical properties were investigated over a broad temperature range. The stress–strain curves of the BcF-CAs were obtained by keeping them in a temperature range of 900 °C–1200 °C for 30 min. As expected, the resulting curves differed imperceptibly (Fig. 2e)[34]. Remarkably, after calcination at 1300 °C, the BcF-CAs exhibited a tensile strength of 88.58 MPa, while those calcined at 1400 °C had a lower strength (15.27 MPa) (Fig. 2f). Additionally, the BcF-CAs retained over 90% of their initial strength and nearly the same Young's modulus and maximum stress even at 1200 °C (Fig. 2g and Supplementary Fig. 20), demonstrating the thermal stability of the structure. The BcF-CAs had reversible strain even upon exposure to a ~1200 °C butane blow torch flame or in liquid nitrogen (−196 °C), and no loss of strength or stiffness was observed (Fig. 2h and Supplementary Fig. 21). Therefore, the temperature-invariant tensile properties of BcF-CAs from −196 °C to 1200 °C were demonstrated.

### Evaluation of temperature-invariant superelastic performance

After the tensile mechanical properties of the BcF-CAs were analyzed, the remarkable compressive performance was then investigated[30]. First, an extensive series of compression tests were conducted by Dynamic Mechanical Analysis (DMA). As expected, the stress-strain ($\sigma$-$\varepsilon$) curves from compressive testing show the maximum compressive stress was up to 156.47 kPa at 80% strain (Fig. 3a)[35]. Subsequently, a comprehensive evaluation of the behavior of the BcF-CAs upon 1000 loading-unloading cycles was conducted. Hysteresis curves of BcF-CAs for 1000 cycles exhibited complete recovery to their original level with slight plastic deformation, confirming the good mechanical resilience and elasticity (Fig. 3b). Remarkably, even after rigorous testing, the BcF-CA exhibited a nearly constant energy loss coefficient and Young's modulus from 10 cycles to 1000 cycles, and retaining ~70% of the original maximum stress (Fig. 3c), demonstrating their structural robustness[37–40].

The macroscopic compressive mechanical behaviors of the Bouligand structures were also simulated by the FE method (Fig. 3d). The results indicated that the small $\alpha$ enhanced the ability to tolerate large deformations. Owing to as the result of the decreased $\alpha$, the corresponding increase in $D$ and fibers ratio occurred, thereby inducing greater energy dissipation. Furthermore, the elastic properties exhibited dependence on $\alpha$, whereby an arrangement with a small $\alpha$ induced a gradual variation in in-plane stiffness and was expected to reduce interlaminar shear stresses - a critical factor contributing to dislamination[30]. Moreover, for small values of $\alpha$, strain augmentation occurred, which culminated in a twisted crack surface that accentuated the surface area per unit crack length along the direction of crack propagation[30].

Microscopically, the superelastic characteristic of the BcF-CAs can be principally attributed to the distinctive helical hierarchical microstructure of their Bouligand array (Supplementary Fig. 22), which was systematically simulated by the FE method[41]. As shown in Fig. 3e, the

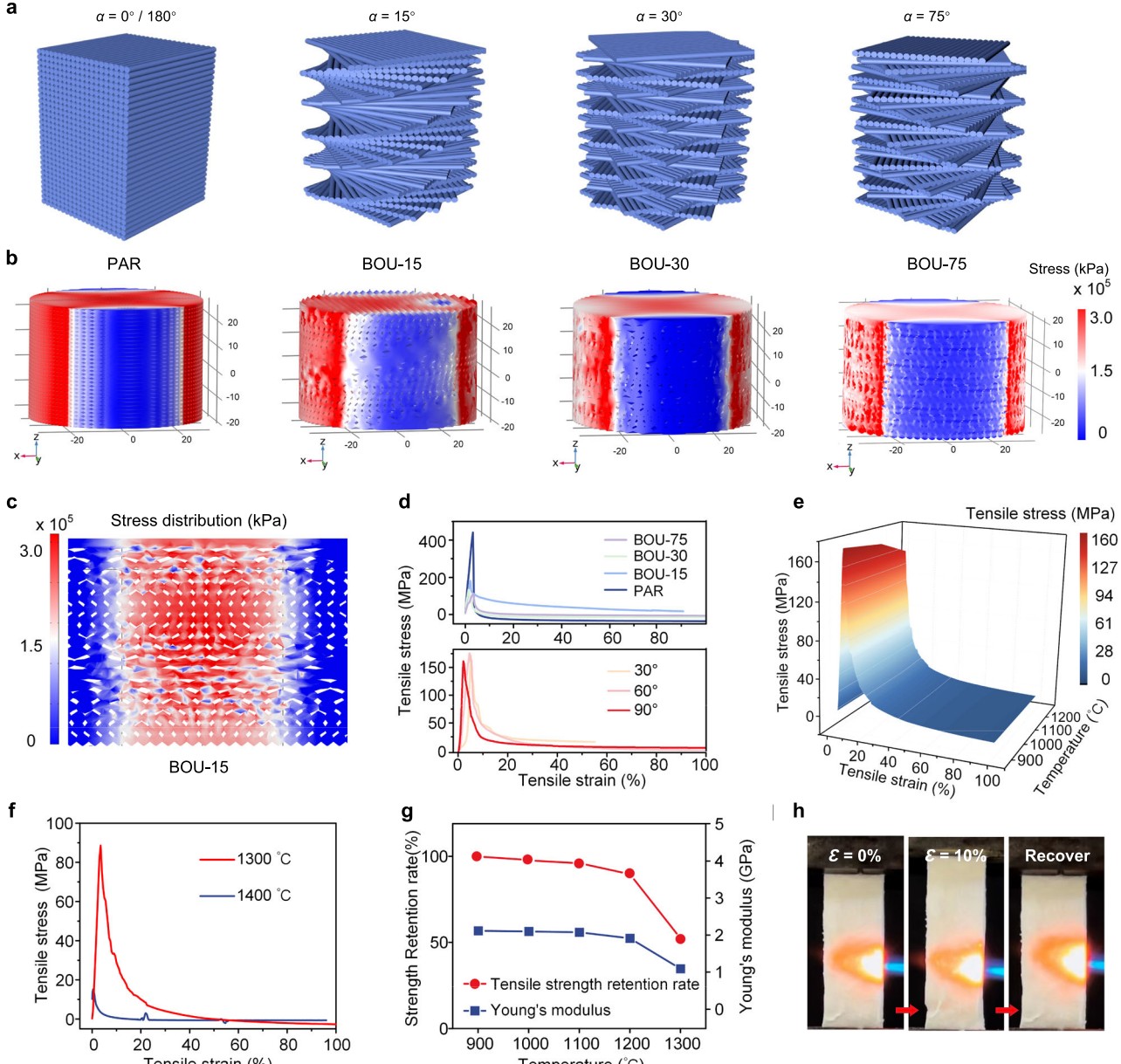

**Fig. 2 | Tensile properties of BcF-CAs. a** The Schematic diagrams of the Bouligand fibers array models with different $\alpha$ values. **b** Simulations through COMSOL Multiphysics showing the tensile stress nephograms of arrays (PAR, BOU-15, BOU-30, BOU-75) under the same tensile load. **c** Stress nephograms of BOU-15 depict the changes in axial stress in the $y$–$z$ plane. **d** Stress–strain curves corresponding to BcF-CAs (BOU-15) with different $\alpha$ (top) and different directions of the force applied (bottom). **e** 3D surface plots of stress versus strain and heat treatment temperature (900-1200 °C) for 30 min. **f** Tensile stress–strain curves of BcF-CAs maintained in a high-temperature environment (1300 °C and 1400 °C for 30 min). **g** Strength retention and Young's modulus as functions of temperature. **h** Tensile recovery process of BcF-CAs in a butane blow torch flame.

helical fiber layers underwent extensive out-of-plane deformations during compression, but absorbed small strains in the plane, which enabled them to tolerate large deformations without structural failure[1–3,41]. Additionally, the elastic strain energy density (ESED) of arcuate-like fiber layers was simulated as shown in Supplementary Fig. 23. The thin hysteresis loops resulted from the effect combined of "rubber" bonding between macro/nanofibers, tough fiber units, and reduced friction between layers or fibers due to the very thin fiber layers and sufficient deformation space. In this regard, this hierarchical structure of the Bouligand array, containing both soft and strong fiber constituent blocks, was very advantageous for good elastic recovery and energy loss under compressive loading[39–42]. Surprisingly, a sequence of real-time photograph captured by a camera showed that when a steel ball (8.47 g) was dropped on a BcF-CA at a high bounce

velocity (910 mm s⁻¹) (Fig. 3f), it rebounded, which clearly demonstrated the superelastic performance of BcF-CAs[1–3].

Similarly, the mechanical stability of BcF-CAs under compression loading in high temperatures environment was studied. The BcF-CAs were annealed at temperatures of 1000 °C, 1100 °C and 1200 °C for 1 h. Subsequently, the compressive loading-unloading behavior was evaluated uniaxially with a compression stain of 80%, and the result showed that there was no significant difference over the entire temperature range (Fig. 3g and Supplementary Fig. 24). In addition, the elasticity of the BcF-CAs at extreme temperatures was examined by compression testing under an intense butane blow torch flame, which reached a temperature of approximately 1200 °C. Impressively, during the whole compressive procedure, no ignition or structural degradation was detected (insets in Fig. 3g)[43]. Moreover, as shown in Fig. 3h,

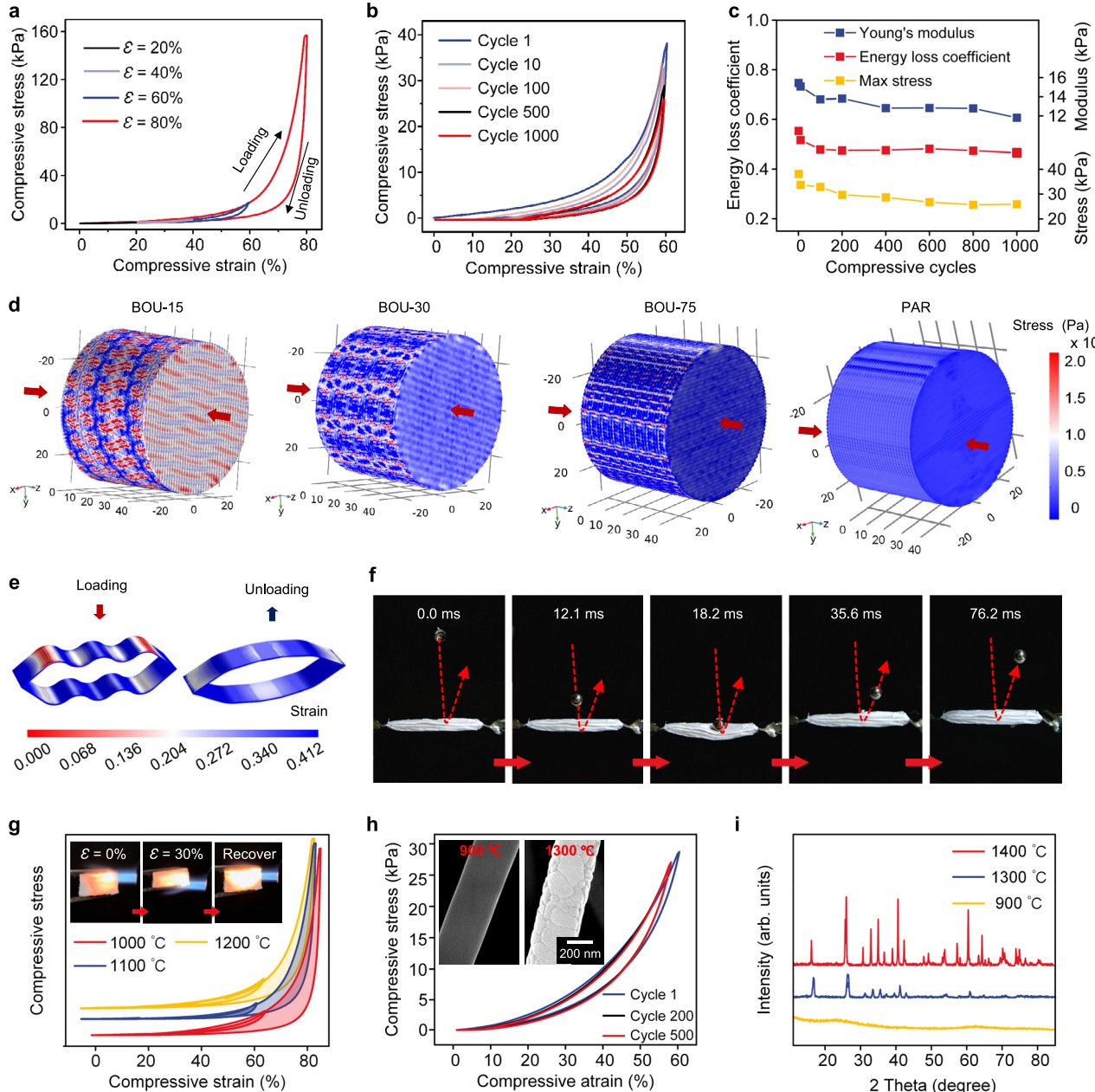

**Fig. 3 | Compressive properties of BcF-CAs. a** Compressive stress–strain curve of the BcF-CAs. **b** Compression test at 60% strain for 1000 loading-unloading cycles. **c** Energy loss coefficient, maximum stress and Young's modulus as a function of the number of compression cycles. **d** Compressive stress nephograms of BOU-15, BOU-30, BOU-75 and PAR models; the arrows indicate the direction of the force. **e** Strain profiles of the arched wall during compression and release process. **f** A series of real-time photographs showing that a BcF-CA was able to rebound a steel ball with a high speed. **g** Compressive stress–strain curves of BcF-CAs exposed to high-temperature environments (1000 °C, 1100 °C and 1200 °C) for 1 h; (insets) Compression–release process of BcF-CAs in a butane blowtorch flame. **h** Multicycles compression test at 60% strain after heat treatment at 1200 °C for 1 h; (insets) SEM images of BcF-CAs after exposure to 900 °C and 1300 °C for 1 h. **i** XRD spectra of the mullite nanofibers.

even after being subjected to 1200 °C for 1 h, the BcF-CAs were able to withstand 500 compression-release cycles with a 60% strain and showed no appreciable plastic deformation, while the maximum compressive stress remaining at 90% of the first compression stage (Supplementary Fig. 25). Meanwhile, this consistent energy loss coefficient (~ 0.22) in the cyclic compression testing demonstrated the structural stability of BcF-CAs. More importantly, the maximum stress and Young's modulus remained constant after the compression test, highlighting the compressive elasticity and fatigue resistance of the BcF-CAs[1–3,44]. Furthermore, the BcF-CAs could maintain their previous morphology after compressing in liquid nitrogen (−196 °C)

(Supplementary Fig. 26). After multiple compression cycles, the BcF-CAs could return to their original geometry, and no apparent breakage was found. Hence, the temperature-independent compressibility, strength and flexibility of the BcF-CAs materials from −196 to 1200 °C were proven.

To obtain more in-depth insight into their thermal stability, the BcF-CAs were further characterized by X-ray diffraction (XRD). Even after prolonged annealing at temperatures as high as 1200 °C, the characteristic peaks of the mullite fibers remained consistent with those of single mullite crystal, highlighting the superior thermal stability under extreme conditions (Fig. 3i and Supplementary Fig. 27)[8–10].

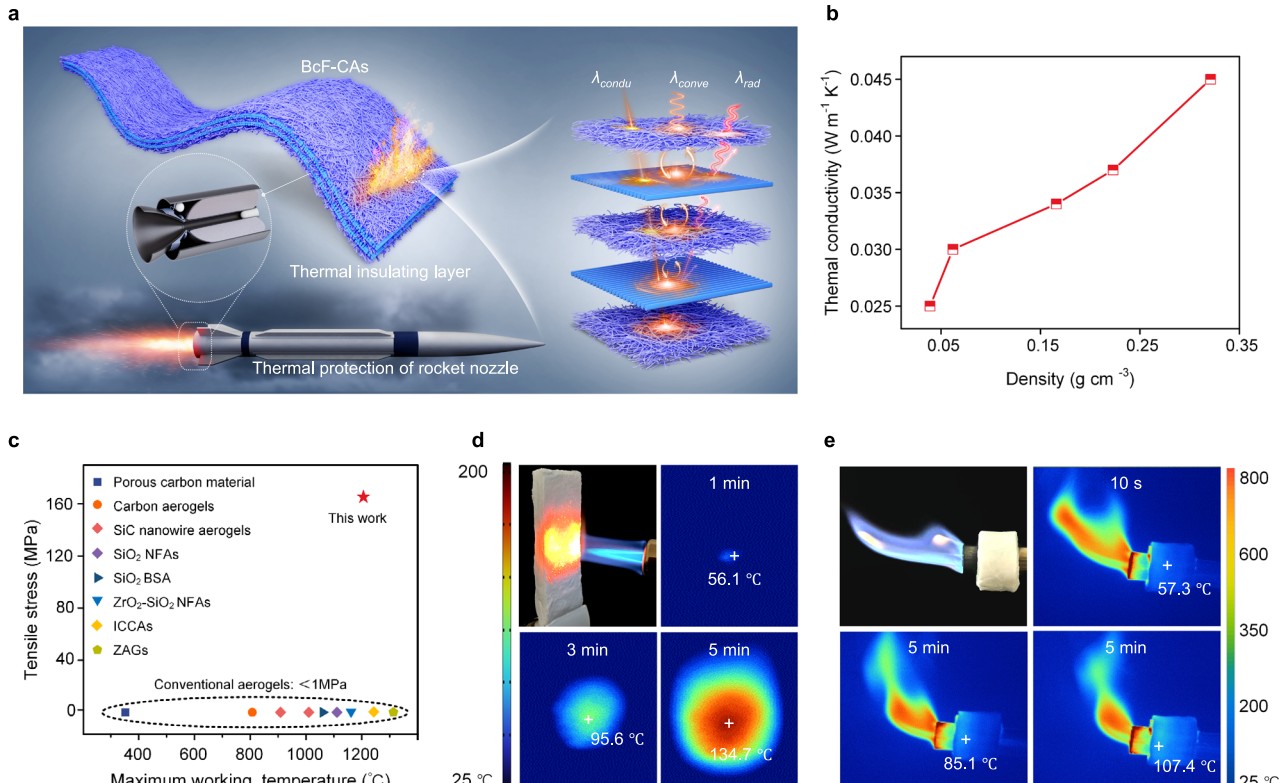

**Fig. 4 | Thermal insulation properties of BcF-CAs. a** Thermal conduction mechanism in the BcF-CAs. **b** Thermal conductivity of BcF-CAs as a function of density. **c** Comparing the thermal conductivity and maximum working temperature of various aerogel-like materials. **d** Optical and infrared images of the BcF-CAs exposed to a butane blow torch for 5 min. **e** Infrared images of a BcF-CA-protected butane nozzle ignited for 5 min.

These results were consistent with the SEM images of the surface morphology. After the BcF-CAs were calcined at 1300 °C, overlarge crystalline grains were formed, resulting in the brittle fracture of the mullite fibers (insets in Fig. 3h and Supplementary Fig. 28)[8].

### Fire resistance and thermal insulation applications

The assembled biomimetic BcF-CAs exhibited superior high-temperature mechanical performance and thermal stability, making them an appealing for use as insulators for a wide range of applications, for instance, as an insulation layer in the aerospace industry (Fig. 4a)[1,11,45]. Therefore, the ability of the BcF-CAs to protect aero-engine from damage at high temperatures was investigated.

The essential features of materials that can be used in thermal shock environments are low thermal conductivity ($\lambda_t$) and low density[1–3,46,47]. Theoretically, the $\lambda_t$ of materials depends on (i) solid ($\lambda_{solid}$) and air ($\lambda_{air}$) thermal conduction ($\lambda_{condu}$), (ii) thermal convection ($\lambda_{conve}$), and (iii) heat radiation ($\lambda_{rad}$)[8–10,48].

$$\lambda_t = \lambda_{condu} + \lambda_{conve} + \lambda_{rad} \qquad (1)$$

$$\lambda_{conve} = \lambda_{solid} + \lambda_{air} \qquad (2)$$

For the BcF-CAs, $\lambda_{conve}$ was severely restricted due to the blocking of air in individual pore[43]. At high porosity, $\lambda_{conve}$ is substantially decreased because the thermal conductivity of air is much less than that of solid materials[8–10,48]. In addition, a hierarchical structure (layer by layer) is deemed to be more beneficial for reflection than a random laminar structure, generating a "multiple reflection effect" at a consistent angle of incidence[48]. Therefore, the unique design of the hierarchical Bouligand chiral array in combination with the ultralight character of aerogel materials resulted in the low $\lambda_t$ of BcF-CAs[1–3,10,49].

As expected, the BcF-CAs had $\lambda_t$ of 0.037 W m$^{-1}$ K$^{-1}$ with a density of 0.223 g cm$^{-3}$ at room temperature (Fig. 4b). In the case where the density was raised to 0.321 g cm$^{-3}$, the value of $\lambda_t$ was enhanced to 0.045 W m$^{-1}$ K$^{-1}$, which was a consequence of the reduced porosity produced by the higher content of macrofibers. It is worth noting that the BcF-CAs showed a substantially diminished $\lambda_t$, measuring about 0.076 W m$^{-1}$ K$^{-1}$ at 300 °C, indicating the exceptional thermal insulation capabilities of BcF-CAs even under extreme conditions and accentuating their potential for application as highly efficient thermal insulating materials.

This low thermal conductivity, along with temperature-independent mechanical qualities, enabled the BcF-CAs to be employed at temperatures above those that typical insulation materials cannot withstand[12]: (i) Most state-of-the-art ceramic aerogel materials, such as SiO$_2$ nanofibrous aerogels[1,8,50–52], ZrO$_2$-SiO$_2$ nanofibrous aerogels[40], SiC nanorod aerogels[53,54], and BN aerogels, can withstand relatively high working temperatures (1100–1300 °C) but have low tensile strength (<1 MPa). (ii) In ontrast to conventional thermal insulation materials, the BcF-CAs possessed tensile properties with maximum working temperature up to ~1200 °C (Fig. 4c and Supplementary Table 1).

As a proof-of-concept for the use of the BcF-CAs as a promising alternative to traditional thermal insulation materials, their structural-mechanical responses investigated in under demanding high-temperature environments[1,54]. An ultra-high temperature testing apparatus was devised in this laboratory to mimic the thermal operating conditions of interest. The BcF-CA was heated and exposed to flames, and the dynamic temperature distribution was monitored using an infrared camera[55]. Firstly, the BcF-CA (15 mm in thickness) underwent direct exposure to a butane flame (~1200 °C), during which time-dependent thermal infrared images were captured to dynamically observe its thermal insulation performance. The BcF-CAs

demonstrated thermal insulating properties in the presence of the intense high-temperature flame, maintaining a low backside temperature of 134.7 °C even after 5 min of continuous heating (Fig. 4d). Following this, a similar approach was employed with a BcF-CA sample exhibiting an 8 mm thickness and a density of 0.21 g/cm³, encasing it around the nozzle of the butane blow lamp (Fig. 4e and Supplementary Fig. 29). While the temperature of the nozzle swiftly surpassed the equipment limit (800 °C) after 5 min, the temperature of the BcF-CAs experienced gradual and restrained escalation, ultimately peaking at approximately 107.4 °C. Notably, after being exposed to high temperatures for 5 min, the temperature outside the BcF-CAs remained almost steadfast, elucidating the superiority of the BcF-CAs as highly desirable thermal insulation materials. These findings revealed that the BcF-CAs provided a high-performance combination of thermal insulation performance and a high degree of architectural stability[25,56].

## Discussion

In this study, a biomimetic compositional and structural engineering strategy was developed via the combination of Bouligand chiral template assembly and subsequent freeze-shaping treatment, which were applied to synergistically form BcF-CAs at the micro/nano scale. Remarkably, the biomimetic BcF-CAs contained Bouligand arrays with tunable, anisotropic and superior mechanical properties, including high-strength tensile properties (170.38 MPa) in the direction of the vertical helix fiber array, and superelastic behavior (156.47 kPa at 80% strain) along the helix fiber array direction. Furthermore, even after exposure to calcination at 1200 °C, the BcF-CAs retained 90% of their original tensile strength, rendering them highly suitable for thermal superinsulation applications. The results of this study demonstrate that this strategy holds immense potential for the advancement of ceramic fibrous aerogels, enabling their utilization across various fields such as thermal insulation, aerospace, environmental remediation, biomedical applications and protective clothing. Moreover, given the generality and customizability of the design strategy, this strategy will be generalizable to other 3D structural materials, thereby extending the suitability of these materials to more demanding mechanical workloads.

## Methods

### Materials

Aluminum chloride hexahydrate ($AlCl_3·6H_2O$, 97%, Aladdin), Aluminum isopropoxide (AIP, $Al(C_3H_7O)_3$, 98.5%, Aladdin), Tetraethyl orthosilicate (TEOS, Si $(OC_2H_5)_4$, 98%, Greagent), Oxalic acid (99%, Aladdin), Ethanol (EtOH, 99.7%, Greagent), Polyvinyl epoxy (PEO, $M_n$ = 60 w, Aladdin), ceramic micron filament fibers (3 M), Boric acid (99.5%, Aladdin).

### Preparation of flexible nanofibrous membranes

The mullite nanofibers were synthesized via the sol–gel method combined with an electrospinning process. Three interrelated steps were as follows: Firstly, the mullite sol was prepared by dissolving 21.729 g of $AlCl_3·6H_2O$, 45.954 g of AIP and 0.3 g of oxalic acid in 130 g of water/ TEOS/ EtOH (with a mass ratio of 4.5: 1: 0.9), which was stirred at 25 °C for 12 h. PEO was then introduced into the mullite sol with continuous stirring for an additional 10 h at room temperature to obtain mullite precursor solution. Subsequently, the mullite nanofiber precursors were directly spun via a DEXS-4 spinning machine, with an operating voltage of 25 kV, collection length of 200 mm, and a steady ejection speed of 3 mL h⁻¹ at a humidity of 35 ± 2% and temperature of 25 ± 2 °C. Finally, the green mullite nanofibers were immediately vacuum dried at 100 °C for 1 h and calcined in air at 1000 °C for 1 h with a muffle furnace, allowing the polymer to complete pyrolysis and production of mullite.

### Preparation of AlBSi sol

The first choice for the high-temperature binder was AlBSi sol. The AlBSi sol was synthesized by stirring at 25 °C for 6 h at an Al/Si/B molar ratio of 2:5:2 (0.39 g $AlCl_3$, 1.52 g TEOS, 0.18 g $H_3BO_3$, 0.016 g $H_3PO_4$, 20 ml of water).

### Preparation of BcF-CAs

The ceramic micron filament fibers were cut into staple fibers of the same length (with a length of $h$) and used as the mechanically reinforced phase of the nanofibrous aerogels; the mullite nanofiber membranes were cut into squares with side length $h$. Both ceramic nanofiber membranes and micron fibers were impregnated in AlBSi aqueous solution at different contents for 2 h. The immersed macrofibers and nanofibers were arranged in layers with a specific helix angle (0°, 15°, 30°, 75°) and then freeze-dried for ~48 h. Then, 3D stacking fibrous aerogels with Bouligand structure were annealed at 900 °C for 1 h in flowing air to obtain the bonded BcF-CAs.

### Characterization

The microstructures of BcF-CAs and mantis shrimp were characterized by FE-SEM (S-4800) and TEM (JEM-2100F). The composition of the AlBSi ceramic was characterized by EDS (Bruker Quantax 400) and XPS (Escalab 250Xi). The chemical structure of BcF-CAs after annealing in a high-temperature condition was determined by XRD (Bruker D8, Bruker). The tensile mechanical tests were measured by an Instron 34 TM-5 universal machine. The compression mechanical tests were performed using a DMA instrument (TA-Q850). The thermal conductivity of BcF-CAs at a range of densities was characterized using a hot disk instrument (TPS2500S, Switzerland). An infrared thermal camera (Fluke TiX560) was used to take the infrared images of the BcF-CAs.

## Data availability

All data generated in this study are provided in the Source Data file. Source data are provided with this paper.

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

## Acknowledgements

This work is supported by the Ministry of Science and Technology of China (No. 2021YFE0105100), the National Natural Science Foundation of China (No. 51925302), the Fok Ying Tong Education Foundation for Young Teachers in the Higher Education Institutions of China (No. 171065), the Science and Technology Commission of Shanghai Municipality (No. 22dz1200102), Fundamental Research Funds for the Central Universities and Graduate Student Innovation Fund of Donghua University (No. CUSF-DH-D-2022025).

## Author contributions

Y.S. and B.D. conceived and designed the study. L.C. and H.W. organized the data. H.W. wrote the manuscript. J.Y. supervised the project. All authors discussed the results and reviewed the manuscript.

## Competing interests

The authors declare no competing interests.
