## [Peer Review File · Nature Communications]

Biomimetic Bouligand Chiral Fibers Array Enables Strong and Superelastic Ceramic AerogelsREVIEWER COMMENTS

Reviewer #1 (Remarks to the Author):

In this work, strong and superelastic ceramic aerogels were fabricated by utilizing the Bouligand structure. The idea for preparing the material is fresh and interesting as well as the outstanding result. However, the structure and writing style of the paper is a bit difficult for readers to follow. There are some information and variables that are not introduced well for full understanding.

1. In the Abstract, could the author include the density and insulation properties of the developed material since it aims for insulation and aerospace technology?
2. In the Introduction, data should be added to support opinions, e.g., "Ceramic aerogels exhibit an extraordinary combination of low thermal conductivity, high stability in extreme environments,...", "...It has been shown that the transition from longitudinally staggered array to Bouligand chiral array leads to a monotonic decrease in anisotropic mechanical properties..."
3. Could the author introduce some previous research working with the Bouligand structure to illustrate how good it is?
4. What are the advantages of combining both macro- and nano-scale components into ceramic aerogels? Where is the idea come from?
5. The paragraphs after the subheading "Fabrication and Bouligand chiral architecture of BcF-CAs" look a bit long and difficult to understand. Could the author shorten them and make them clearer? In addition, it could be better to have a diagram to depict how BcF-CAs are fabricated from preparing nanofibrous membranes to the final material.
6. Letter "a" and "α" are not used consistently in the Figures and the paragraphs even though they have the same meaning. The ϕ symbol should be illustrated in Fig. 1b.
7. Temperature and the surface of the material before and after applying the butane blowtorch should be shown in Fig. 1m to support the idea "...the BcF-CA maintained their shape and structure even under such extreme conditions...".
8. The author should investigate the effect of AIBSi sol on the mechanical properties of the material to choose 5 wt% as an appropriate concentration.
9. Could the author show how to determine the thickness of the bonding layer?
10. Could the author give more description for Fig. 2b?
11. Fig. 11d does not look quite obvious.
12. As the maximum tensile stress is around 170.38 MPa, however, the axis value for the stress in Fig .2g used the unit of GPa, which is difficult to see how the stress change over the temperature range.
13. what is the micro-fiber content and ratio of fibers mentioned in the manuscript? How to control them when preparing the material?
14. The l symbol is used in the manuscript with the meaning of the number of layers but meaning the side length in the fabrication process (supplementary file).
15. What is the strength retention and energy loss coefficient? How to determine them?

16. Did the author demonstrate the temperature-invariant stretchability tensile properties of BcF-CAs at around -196 °C?

17. How about the tensile and compression strength of continuous blocks constructed by only microfibers and nanofibers compared to anisotropic Bouligand ceramic aerogels?

18. There are typos: a significant compression stain of 80% (page 14), Discussion heading should be Conclusion.

19. Which sample is mentioned in Fig. 3a-c?

20. Could the author double-check Young's modulus in Fig. 3c? What is the strain used to determine the stress in that figure?

21. Does changing the torsion angle affect the density and thermal conductivity of the material?

22. Please provide reference and thermal conductivity (if possible) for Supplementary Table 1.

Reviewer #2 (Remarks to the Author):

In the present manuscript the authors demonstrate the "Bouligand" (or twisted) structure that are seen in nature is suitable for strengthening the fiber-based porous materials. The idea is interesting and rather easy to be realized, and the tensile strength of the freeze-dried aerogel-like material shows impressively high. In my opinion, however, in addition to the enhancement of mechanical strength, the authors have to report the thermal conductivity at high temperatures since the authors mention applications in the aerospace industry. To gain enough novelty to be accepted to Nature Communications, this reviewer requests to clarify this point (comment 5 below), together with some other revisions.

1. The preparation process should be briefly written in the main text so that readers can understand what the materials are like. Especially for understanding how the angles were made between the layers and what is the AIBSi sol and its role.

2. Why were the two precursors of aluminum chloride and aluminum isopropoxide used?

3. Please specify the role of the AIBSi sol. In Fig. 1i-k, it is described a surface layer is formed on the nanofibers, and silicate bonds (X-O-Si) forms at elevated temperatures. However, I cannot believe the firm bonds form in-between the 1-nm layers, so it is desired to show SEM images that prove the formation of the crosslinks. By the way, is there a peak of B in XPS? It seems to be so small that not possible to see in the survey spectrum.

4. In page 17, "For the BcF-CAs, λ_{conv} is severely restricted due to the blocking of air in single micropores", but there is no discussion on the micropores so far. Instead from SEM images, the pores are in the micrometer order so they should be called as macropores.

5. The fatal drawback of this manuscript is there is no report on the thermal conductivity at elevated temperatures. The thermal conductivity values at room temperature are reported, but the novelty of this material is mechanical strength at high temperatures like 1200 deg C, and the authors mention the use as an insulation layer in the aerospace industry (Fig. 4a). Therefore, it is needed to show the thermal conductivity values at such high temperatures, without which readers cannot judge this material is suitable for such applications or not. In addition, since the material loses the strength above 1300 deg C, it must be used at 1200 deg C or lower. The authors should verify that the 1200 deg C is high enough for those applications or not. I am not sure how high the temperature is in the exhaust of the rocket engines. Also, the high-temperature thermal conductivity must be lower

compared to other insulation materials, and the authors have to prove the value is low enough for such applications.

Point-by-point responses to the comments

***Response summary:** We sincerely thank the reviewer for the invaluable comments on our manuscript; the insightful comments have played a crucial role in enhancing the overall quality of the work. In response to their suggestions, we have diligently revised the manuscript and double-checked similar deficiencies throughout the manuscript. We earnestly hope that the reviewers find our responses satisfactory and convincing. Detailed responses to each comment are provided below.*

Comments from Reviewer #1:

In this work, strong and superelastic ceramic aerogels were fabricated by utilizing the Bouligand structure. The idea for preparing the material is fresh and interesting as well as the outstanding result. However, the structure and writing style of the paper is a bit difficult for readers to follow. There are information and variables that are not introduced well for full understanding.

***Response:** Thank the reviewer very much for the positive and constructive comments. We highly appreciate the reviewer's acknowledgment of the novelty aspect of our work.*

*We apologize for the **structure problems** of our manuscript. Actually, in this manuscript, we developed Bouligand-chiral and fibrous structured ceramic aerogels (BcF-CA) with strong, superelastic, and fatigue-resistance properties for thermal insulation applications. **The whole work includes four parts:***

- i) Fabrication and Bouligand chiral architecture of BcF-CAs;*
- ii) Evaluation of temperature-invariant tensile performance;*
- iii) Evaluation of temperature-invariant superelastic performance;*
- iv) Fire resistance and thermal insulation applications.*

***In the first section,** the inspiration and preparation process of the BcF-CAs material were elaborated in detail. **Subsequently, in the second section,** the finite element (FE) method was employed to thoroughly simulate the tensile properties of BcF-CAs under varying torsion angles. Additionally, the high-temperature tensile properties of the BcF-CAs were also examined. **The third section** focused on studying the impact of twisting angles on the compressive properties of BcF-CAs and their ability to withstand extreme environments ranging from -196 °C to 1200 °C. **Finally,** the thermal insulation properties of BcF-CAs were evaluated. The assembled BcF-CAs exhibited exceptional high-temperature mechanical and thermal stability, indicating their potential as a desirable insulation material in various fields, including insulation layers in the aerospace industry.*

Furthermore, we sincerely apologize for any previous writing issues encountered in the manuscript. To address this concern, the manuscript has undergone a thorough revision by native English speakers, ensuring improved clarity and coherence. Subsequently, all authors have diligently reviewed and refined the manuscript before submission.

1. In the Abstract, could the author include the density and insulation properties of the developed material since it aims for insulation and aerospace technology?

Response: *Thank the reviewer for the valuable and constructive comments, the insightful feedback is highly beneficial in refining and improving our work.*

*As suggested by the reviewer, we have described the density and insulation properties of the BcF-CAs in the “**Abstract**” in detail in the revised Manuscript.*

Abstract:

“Ceramic aerogels are often used when thermal insulation materials are desired; however, they are still plagued by poor mechanical stability under significant thermal shock. Here, inspired by the dactyl clubs of mantis shrimps found in nature, which form by directed assembly into hierarchical, chiral, Bouligand (twisted plywood) structures with improved mechanical properties, we present a compositional and structural engineering strategy to develop strong, superelastic, and fatigue resistance ceramic aerogels with chiral fibers arrays resembling Bouligand architecture. Benefiting from the stress dissipation, crack torsion, and mechanical reinforcement of micro-/nano-scale Bouligand arrays, the tensile strength (170.38 MPa) of these aerogels is between one and two orders of magnitude greater than that of state-of-the-art nanofibrous aerogels. In addition, the developed aerogels feature exceedingly low density (0.223 g cm⁻³), low thermal conductivity (0.037 W m⁻¹ K⁻¹), exceptional compressive properties with rapid recovery from 80% strain, and excellent thermal stability up to 1200 °C, making them ideal for thermal superinsulation applications. This technology holds great promise for designing and assembling aerogel architectures without the typical limitations of traditional fabrication methods.”

2. In the Introduction, data should be added to support opinions, e.g., “Ceramic aerogels exhibit an extraordinary combination of low thermal conductivity, high stability in extreme environments,...”, “...It has been shown that the transition from longitudinally staggered array to Bouligand chiral array leads to a monotonic decrease in anisotropic mechanical properties...”

Response: *We sincerely thank the reviewer for the valuable comments; the comments are constructive in improving our manuscript.*

As suggested by the reviewer, we have added data mentioned by the reviewer to support opinions in the manuscript (page 3, line 45; page 3, line 87).

Page 3, line 45, “Ceramic aerogels exhibit an extraordinary combination of low thermal conductivity (from 0.012 W m⁻¹ K⁻¹ to 0.033 W m⁻¹ K⁻¹ in the air), good stability in extreme environments (-196 ~ 1200 °C), ...”,

Page 4, line 87, “...it has been shown that the transition from a longitudinally staggered array to a Bouligand chiral array leads to a monotonic decrease in anisotropic mechanical properties, i.e., the anisotropy ratio of Young’s modulus changes from 4.25 to 1.25...”

Moreover, we have double-checked similar deficiencies throughout the manuscript.

3. Could the author introduce some previous research working with the Bouligand structure to illustrate how good it is?

Response: We sincerely thank the reviewer for the valuable comments; the comments are constructive in improving our manuscript.

As suggested by the reviewer, we have provided previous research on Bouligand structure in the revised Manuscript (Page 4, line 81).

Page 4, line 81:

“Previous studies have shown that the Bouligand chiral conformation provides superior energy absorption, effective stress transfer, and the ability to inhibit crack propagation by twisting and reorienting ordered nanofibers under external loads, thereby endowing materials with outstanding mechanical properties including toughness, fracture resistance, and impact resistance.^{1,2} In addition, it has been proved that the transition from a longitudinally staggered array to a Bouligand chiral array leads to a monotonic decrease in anisotropic mechanical properties, e.g., the anisotropy ratio of Young’s modulus changes from 4.25 to 1.25.^{3,4} Other studies have demonstrated that materials with a Bouligand array exhibit band gaps at frequencies associated with the impact stress pulse and therefore confer wave-filtering property.⁵ Collectively, these studies support the integration of microfibers to reinforce conventional ceramic nanofibrous aerogels, form architectures similar to Bouligand chiral arrays, and customize strong mechanical performance and other functionalities in ceramic aerogel.⁶⁻⁹”

4. What are the advantages of combining both macro- and nano-scale components into ceramic aerogels? Where is the idea come from?

Response: We sincerely thank the reviewer for the valuable comments; the comments are constructive in improving our manuscript.

Ceramic aerogel materials exhibit the advantages of **low density, low thermal conductivity, high temperature resistance, and superior thermal chemical stability**¹⁰; nevertheless, the actual utilization of ceramic aerogels **remains restricted** owing to their **inherent hardness, brittleness, and incompressibility**. In order to address these limitations and enhance the mechanical properties of the ceramic aerogels, implementing ceramic nanofibers as fundamental building blocks to fabricate nanofibrous aerogels has proven to be **a more efficacious approach**. Such nanofibrous aerogels exhibit improved flexibility and elasticity, **effectively overcoming the brittleness** associated with conventional ceramic aerogels¹¹⁻¹⁴. However, due to the **macroscopic mechanical limitations of the nanocomponents**, these aerogels exhibit **relatively low mechanical stress (<1 MPa)** and are incapable of withstanding the harsh mechanical and thermal flow shock encountered in practical applications.¹⁴

Ceramic microfibers, characterized by exceptional mechanical properties such as tensile strength ranging from **1.75 GPa to 3.1 GPa**,¹⁵⁻¹⁷ coupled with high-temperature stability **surpassing 1200 °C**, have been extensively employed as **reinforcing phases** in fabricating **ceramic matrix composites**. Incorporating these microfibers has shown a remarkable capability to significantly boost the tensile strength (159-352 MPa) and flexural strength (159-352 MPa) of the resulting materials.¹⁸⁻²² **Regrettably, the ultra-high density and extreme reliance on the fibers orientations** in determining material mechanics (with the highest mechanical properties observed along the axial direction of the fibers) **have severely limited their further expansion** in the field of high-temperature thermal insulation.

Traditional Chinese Taoism emphasizes the balance of the five elements, particularly the concept of **“Yin and Yang, rigidity and flexibility”**. This philosophical notion of **balancing opposing forces** has relevance in scientific endeavors, particularly in advancing our understanding of materials. In material science, achieving ultra-high strength necessitates the utilization of rigid components. However, a material composed **solely of “rigid”** elements would become brittle and heavy, impeding its practicality. Conversely, **“flexibility”** provides inherent lightness and pliability, but compromises macroscopic mechanics such as tensile and compressive strength **if only “flexibility” elements are used**.

In our work, for example, ceramic microfiber, known for its ultra-high strength, constitutes the **“rigid”** component. It exhibits remarkable properties, but the materials with only microfiber component fall short in **brittleness and weight** (2.3~2.6 g/cm³). Meanwhile, ceramic nanofibers possess inherent **lightness and “flexibility”**, however, the materials of only nanofiber component exhibit inferior macroscopic mechanical characteristics with tensile strength **below 1 MP**. **Therefore, by capitalizing on the**

dichotomy of “rigidity” and “flexibility”, we integrate these two types of fibers to maximize their collective mechanical advantages.

In addition, nature is a rich source of inspiration. One example is the mantis shrimp, an appealing but deadly creature. The complex anatomy of the mantis shrimp's claws, consisting of Bouligand chiral stacking of the reinforced chitin protein fibrils, coupled with their exceptional speed, enable them to breach the defenses of even the most resilient prey such as mollusks and crabs.²³ Motivated by these above, specifically, we propose an approach to utilize ceramic microfibers and nanofibers as bicomponent building blocks, combined with chiral assembly resembling the biomimetic Bouligand helix structure, to maximize the mechanical advantages of micro/nanofibers. Remarkably, we successfully developed ceramic micro/nanofiber aerogels, demonstrating a significantly reduced bulk density of only 0.215 g cm^{-3} while maintaining high-temperature resistance and low thermal conductivity. The exceptional properties of these aerogels position them as prime candidates in high-temperature insulation applications, encompassing aerospace, deep-sea exploration, and military industries.

5. The paragraphs after the subheading “Fabrication and Bouligand chiral architecture of BcF-CAs” look a bit long and difficult to understand. Could the author shorten them and make them clearer? In addition, it could be better to have a diagram to depict how BcF-CAs are fabricated from preparing nanofibrous membranes to the final material.

Response: We sincerely thank the reviewer for the valuable comments, which are very helpful in improving our manuscript.

First, as suggested by the reviewer, we have carefully revised the paragraphs mentioned by the reviewer in the revised Manuscript (Page 5, line 108).

Page 5, line 108:

“The mantis shrimp, an appealing but deadly creature. The complex anatomy of the mantis shrimp's claws, consisting of Bouligand chiral stacking of the reinforced chitin protein fibrils, coupled with their exceptional speed, enable them to breach the defenses of even the most resilient prey such as mollusks and crabs. Typically, the unique Bouligand architecture is characterized by a helical array of fibril lamellae with a twisting angle, as shown in Fig. 1a. The architecture is subjected to a 180° rotation, which can be quantified by α and ϕ along the z-axis direction, where α represents the twisting angle created among two neighboring layers, while ϕ defines the distribution of twisting angle (Fig. 1b). The direction of the fibers aligned with the x-axis is denoted as $\phi = 0^\circ$, meanwhile the fibers twist counterclockwise around the z-axis.”

Page 6, line 118:

“This configuration increases the crack surface area and contributes to the reorientation of the fibers in response to external stresses, such as tensile, flexural, and impact loads²⁹. Meanwhile, the resulting modulus oscillation within the Bouligand geometry is assumed to enhance crack torsion (Supplementary Fig. S1). Incidentally, the Bouligand chiral array result in in-plane isotropic mechanical properties (Fig. 1c-d and Supplementary Fig. S2), which overcome the typical limitations of materials with traditional unidirectional 3D fibrous structure.”

Furthermore, we have provided additional elaboration for Fig. 1b-d to enhance comprehension. As shown in Fig.1b, the Bouligand structure exhibits an intriguing propensity to generate a substantial number of intricately interwoven microcracks characterized by twisting patterns. Meanwhile, ***the crack propagation occurs predominantly within the interfiber domain rather than through the fibers.***^{24,25} Consequently, the progressive development and propagation of these microcracks provide a source of energy dissipation and stress relaxation - ultimately contributing to the exceptional damage tolerance properties of the dactyl club, enabling it to withstand substantial loads without suffering catastrophic failure.

Fig. 1b Structural features and crack propagation path of typical Bouligand chiral array.

In addition, as suggested by the reviewer, we provided a detailed diagram to depict the fabrication process of the BcF-CAs in the revised Supporting Information (Supplementary Fig. 4).

Supplementary Figure 4. Schematic illustration of the manufacturing process of the BcF-CAs.

6. Letter “ α ” and “ α ” are not used consistently in the Figures and the paragraphs even though they have the same meaning. The ϕ symbol should be illustrated in Fig. 1b.

Response: We apologize for our careless mistakes. Thanks for the reviewer’s correction; the comments are very helpful in improving our manuscript.

(1) In our resubmitted manuscript, we have corrected all the “ α ” into “ α ”. (Page 5, line 114; Page 12, line 245; Page 14, line 297; Page 14, line 301). Moreover, we have carefully double-checked similar deficiencies throughout the manuscript.

Page 5, line 114, “Specifically, the architecture is subjected to a 180° rotation, which can be quantified by α and ϕ along the z-axis direction, where α represents torsion angle created among two neighboring layers, and ϕ defines the distribution of the twisting angle along the crack propagation path (Fig. 1b).”

Page 12, line 245, Remarkably, smaller α resulted in an increment in the proportion of reinforcing fibers in the direction of tensile loading, thereby enhancing the ductility and toughness of the BcF-CAs to prevent fracture at the macroscopic level (Fig. 2b).”

Page 14, line 297, “The results indicated that the small α (15°) enhanced the ability to tolerate large deformations.”

Page 14, line 301, “...arrangement with a small α induced a gradual variation in in-plane stiffness and was expected to reduce interlaminar shear stresses - a critical factor contributing to delamination...”

(2) As suggested by the reviewer, we have illustrated the ϕ symbol in Fig. 1b in the revised Manuscript. Thanks for the reviewer's correction.

7. Temperature and the surface of the material before and after applying the butane blowtorch should be shown in Fig. 1m to support the idea that “...the BcF-CA maintained their shape and structure even under such extreme conditions...”.

Response: We sincerely thank the reviewer for the valuable comments, which are very helpful in improving our manuscript.

As suggested by the reviewer, we have provided the temperature and the surface of BcF-CA (Fig. 1m) before and after applying the butane blowtorch in the revised Supporting Information (Supplementary Fig. 14).

Fig. 1m The BcF-CAs were exposed to a butane blowtorch, and no damage was observed.

Supplementary Figure 14. The BcF-CAs were exposed to a butane blowtorch before and after the treatment.

8. The author should investigate the effect of AlBSi sol on the mechanical properties of the material to choose 5 wt% as an appropriate concentration.

Response: We greatly thank the reviewer’s professional review work on our manuscript; the comments are constructive in improving our manuscript.

As suggested by the reviewer, we have provided the compressive and tensile mechanical response of BcF-CAs to the concentration gradient of AlBSi sol in the revised Supporting Information. (Supplementary Fig. 9-10).

Supplementary Figure 9. Compressive stress-strain curves of BcF-CAs with AlBSi–0.5 wt%, AlBSi–2 wt%, AlBSi–4 wt%, AlBSi–5 wt%, AlBSi–10 wt%.

Supplementary Figure 10. Tensile stress-strain curves of BcF-CAs with (a) AlBSi–0.5 wt%, (b) AlBSi–2 wt%, (c) AlBSi–4 wt%, (d) AlBSi–5 wt%, (e) AlBSi–8 wt%.

As shown in **Supplementary Fig. 9**, the maximum compressive stress of aerogels stabilized at 160 kPa as the sol concentration increased to 5 wt%. Meanwhile, the tensile stress of aerogels was initially increased and then decreased (**Supplementary Fig. 10**); the maximum stress of 179 MPa was found at 5 wt%. We speculated that a suitable sol content would benefit the robust bonding between the fibers (**Supplementary Fig. 8d** and **Supplementary Fig. 11**). Still, excessively adhesive interfaces would be generated between the fibers and the AlBSi matrices when the sol concentration exceeded a specific value. In the latter case, too much AlBSi sol shrank during drying, but the nanofibers hardly shrank, resulting in tiny cracks induced by strong adhesion, which could accelerate the stress-failure process of BcF-CAs to some extent (**Supplementary Fig. 8e-f**).

Supplementary Figure 8. The surface morphological SEM images of (a) AlBSi–0.5 wt%, (b) AlBSi–2 wt%, (c) AlBSi–4 wt%, (d) AlBSi–5 wt%, (e-f) AlBS–8 wt%.

Supplementary Figure 11. The robust bonding between macro-fibers and nanofibers with AlBSi–5 wt%.

9. Could the author show how to determine the thickness of the bonding layer?

Response: Thank the reviewer for the constructive comments. We apologize for not clearly representing the factors that affect the thickness of the AlBSi bonded layer in the Manuscript, which confuses the reviewer about the design principle.

*We determined the thickness of the bonded layer by **combining SEM and TEM images**. The process of fabricating BcF-CAs, as described in the Q15, involves immersing the micronanofibers in AlBSi sol to deposit a bonding layer on their surface for stable bonding between the adjacent fibers. The thickness of bonding layers mainly depend on the concentration of AlBSi sol and the immersion time. In the Q8, we observed the microscopic morphology of fibers with different sol concentrations, and it can be easily found from the SEM images that the higher the sol concentration, the larger the thickness of the bonding layer on the surface of the fibers.*

In addition, we have supplemented the TEM images of fibers with different immersion times to further observe the variation in the thickness of the bonding layer.

As shown in **Figure 1**, the thickness of the bonded layer on the fiber surface increased with the impregnation time.

Figure 1. HRTEM-EDS images of an individual fiber with immersion time of (a)30 min, (b)2h, (c) 6h.

Meanwhile, as shown in the SEM image (Figure 2), the thickness of the bonding layer on the fiber surface varies at different regions of the aerogel. This can be attributed to the fact that the sol tends to aggregate in the fiber overlap, and thus, the thickness of the bonding layer is thickest at the location where the fibers are bonded.

Figure 2. HRTEM-EDS images of the ceramic bonding layer on the surface of the fiber.

10. Could the author give more description for Fig. 2b?

Response: Thank the reviewer for the constructive comments. We apologize for not presenting the experimental details of Fig. 2b. **In the revised Manuscript**, we have provided corresponding details and discussions (Page 12, line 240).

Page 12, line 240:

“The simulation results clearly showed that as the value of α decreased from 75° to 0° , the maximum stress (i.e., the stress at initial failure) increased (Fig. 2b). Mechanistically, the Bouligand array enabled the fibers to reorientate in response to external stress. The majority of the fibers reorientate along the tensile direction and underwent tensile deformation owing to stretching/sliding mechanisms, while some of the other fibers rotated symmetrically in the direction away from the stress axis. Remarkably, smaller α resulted in an increment in the proportion of reinforcing fibers

in the direction of tensile loading, thereby enhancing the ductility and toughness of the BcF-CAs to prevent fracture at the macroscopic level (Fig. 2b).³⁶”

11. Fig. 11d does not look quite obvious.

Response: Thank the reviewer for the constructive comments. May I understand what the reviewer wants to express is the problem of Fig. 2d?

We apologize for not presenting the details of Fig. 2d.

First, as suggested by the reviewer, we have **revised Fig. 2d** in the revised Manuscript. In addition, we have provided **more detailed descriptions of Fig. 2d** to make it easier for the reader to understand. (Page 12, line 260).

Fig. 2d Stress-strain curves corresponding to BcF-CAs (BOU-15) with different α (top) and different directions of the force applied (bottom).

Page 12, line 260:

“Intuitively, the tensile strength increased as α decreased (Fig. 2d top). Furthermore, the BOU-15 subjected to tensile loading in different directions were further investigated (Supplementary Fig. S18 and Fig. 2d bottom). The results indicated that no significant differences in the tensile strength of BcF-CAs were found even when the force direction was varied. That is, BcF-CAs with chiral helical fiber arrays had tensile properties that were independent of loading directions and fiber orientation, which was generally difficult to achieve in other fiber-based 3D materials. More importantly, as expected, the experimental behavior correlated well with the simulation results.”

Supplementary Figure 18. The BOU-15 was subjected to tensile stress in different directions (30°, 60°, 90°), where the red lines represent the directions of the force applied.

12. As the maximum tensile stress is around 170.38 MPa, however, the axis value for the stress in Fig .2g used the unit of GPa, which is difficult to see how the stress change over the temperature range.

Response: We sincerely thank the reviewer for the valuable comments, the comments are very helpful in improving our manuscript.

As suggested by the reviewer, we have split Fig. 2g into two figures in the revised manuscript and Supporting Information (**Fig. 2g Supplementary Fig. 20**) for ease of understanding.

Fig. 2g Strength retention rate and Young's modulus as a function of temperature

Supplementary Figure 20. The maximum stress of the BcF-CAs as a function of temperature

13. what is the micro-fiber content and ratio of fibers mentioned in the manuscript?
How to control them when preparing the material?

Response: We sincerely thank the reviewer for the valuable comments, the comments are very helpful in improving our manuscript.

(1) what is the micro-fiber content and ratio of fibers mentioned in the manuscript?

Micro-fiber content and ratio of fibers mentioned in the manuscript refers to the mass percentage of microfibers incorporated within the BcF-CAs.

(2) How to control them when preparing the material?

We tested the relationship between material density - mechanical properties - micron fiber content (Supplementary Fig. 19). Considering the need for low density in insulation materials for aerospace applications, and balancing the relationship between the material's mechanics and its density, an optimal ratio of micron fibers was determined. (Page 13, line 271)

Page 13, line 271 :

“Considering the need for ultralight BcF-CAs, the optimum content of microfibers was determined to be 15 wt %.”

Supplementary Figure 19. 3D surface plots of the tensile stress as a function of c and density.

14. The l symbol is used in the manuscript with the meaning of the number of layers but meaning the side length in the fabrication process (supplementary file).

Response: Thanks for the reviewer's careful checks. We are sorry for our carelessness.

As suggested by the reviewer, we have corrected “ l ” to “ h ” in the revised Supporting Information.

Revised Supporting Information:

“Then the ceramic micron filament fibers were cut into staple fibers of the same length (with a length of h) and used as the mechanically reinforced phase of the nanofibrous aerogels; the mullite nanofiber membranes were cut into squares with side length h .”

15. What is the strength retention and energy loss coefficient? How to determine them?

Response: We sincerely thank the reviewer for the valuable comments, the comments are very helpful in improving our manuscript.

(1) What is the strength retention and energy loss coefficient?

■ **Strength Retention**

The Strength Retention in Fig. 2b of the Manuscript refers to the percentage of the strength of the material after being treated at high temperature (900-1300 °C) for 1 h and the strength of the material itself. The value of Strength Retention corresponds to the high-temperature mechanical properties of the material. A significant Strength Retention indicates the material has better resistance to high temperatures.

■ Energy Loss Coefficient

In the case of viscoelastic materials, loading and unloading follow different stress-strain curves, and due to the mismatch between the loading and unloading curves, they form a **closed loop**, which becomes the elastic **hysteresis loop** (Figure 3) The area enclosed by the elastic hysteresis loop is the **dissipated energy per unit volume** of the material (ΔU). (Here, only the loading and unloading of the material under uniaxial stress is considered). However, as shown in **Figure 4 (unclosed-type hysteresis curve)**, it indicates that the material is poorly elastic and has undergone severe plastic deformation.

The **energy dissipation coefficient** is a physical quantity that evaluates the degree of plastic failure, toughness, and deformability of a material. Specifically, it refers to **the ratio of the energy lost to the initial energy ($\Delta U/U$) during compression. For aerogels, the strength, toughness, fatigue resistance, and crack resistance are all enhanced by this energy dissipation mechanism.** Because in the tensile or compression process, the greater the local energy dissipation, the smaller the stress wave propagation distance, which is not conducive to damage, and the material exhibits **lower brittleness and higher fracture resistance.** Meanwhile, as the energy dissipation coefficient increases, the aerogel network recovers rapidly in the uniaxial stretch-relaxation and compression-relaxation modes.¹

Figure 3. Mechanical energy dissipated by loading and unloading aerogel.

Figure 4. Aerogels susceptible to plastic deformation

Furthermore, the stress-strain curves obtained from our experiments with different numbers of cycles are all **nonlinear closed-type smooth hysteresis curves** (Fig. 3b), indicating that our prepared aerogel is a typical material with **high viscoelasticity, high energy dissipation, and can be highly deformable.** These properties are similar to that of rubber. Nevertheless, **Figure 5** represents aerogels that are susceptible to plastic deformation and prone to fatigue under multiple large deformation cycles, respectively.

Fig. 3b Compression test at 60% strain for one thousand cycles.

Figure 5. Aerogels are susceptible to fatigue under multiple cycles of large deformation.

(2) How to determine them?

■ **Strength Retention**

To control (improve) the strength retention of the material, the following points must be considered:

i) Improve the high-temperature resistance of the building blocks; ii) Establish a stable bonding between the building blocks; and iii) design a robust microstructure inside the materials. The biomimetic Bouligand ceramic fibrous aerogels fabricated in our manuscript fulfill these three points (**high-temperature resistant micro/nanofibrous** matrix elements, **stable high-temperature bonding structure** of AlBSi ceramics, and **deformable 3D spatial network** structure of Bouligand), and therefore have ideal strength retention even at extremely high temperature.

■ **Energy dissipation coefficient**

To control (improve) the material with a significant energy dissipation coefficient, it is necessary **to improve the energy dissipation ability of the material during the deformation process.** For BcF-CAs in our manuscript, since the micro/nanofibers are interconnected to form a 3D spatial (Bouligand) chiral network, the **deformation of the three-dimensional network** structure absorbs a certain amount of energy during the compression process, enabling the BcF-CAs to have a high energy dissipation coefficient and thus a good deformation resistance.

16. Did the author demonstrate the temperature-invariant stretchability tensile properties of BcF-CAs at around -196 °C?

Response: We sincerely thank the reviewer for the valuable comments, the comments are very helpful in improving our manuscript.

As suggested by the reviewer, we have **provided a detailed discussion** of the tensile properties of BcF-CAs at around $-196\text{ }^{\circ}\text{C}$ in revised manuscript (**Page 13, Line 282**). Moreover, temperature-invariant stretchability tensile properties of BcF-CAs in liquid nitrogen ($-196\text{ }^{\circ}\text{C}$) **has been provided** in the revised Supporting Information (**Supporting Fig. 21**).

Page 13, Line 282:

“The BcF-CAs had reversible strain even upon exposure to a $\sim 1200\text{ }^{\circ}\text{C}$ butane blow torch flame or in liquid nitrogen ($-196\text{ }^{\circ}\text{C}$), and no loss of strength or stiffness was observed (Fig. 2h and Supplementary Fig. S21). Therefore, the temperature-invariant tensile properties of BcF-CAs from $-196\text{ }^{\circ}\text{C}$ to $1200\text{ }^{\circ}\text{C}$ were demonstrated.”

Supplementary Figure 21. Temperature-invariant stretchability. Tensile and recovery processes in liquid nitrogen.

17. How about the tensile and compression strength of continuous blocks constructed by only microfibers and nanofibers compared to anisotropic Bouligand ceramic aerogels?

Response: We sincerely thank the reviewer for the valuable comments, the comments are very helpful in improving our manuscript.

As suggested by the reviewer, we have investigated the **tensile and compression strength of continuous blocks constructed by only microfibers and nanofibers** and provided more corresponding explanations.

As shown in Figure 6 and Fig. 1i, under the same volume content, **the tensile strength of full-micron fiber aerogel is the highest ($\sim 1\text{ GPa}$), followed by BcF-CAs, and the tensile strength of full-nanofiber aerogel is the lowest (less than 1 MPa)**, which is because the strength of micron fibers is much higher than that of nanofibers and can withstand the more enormous external load, so **the higher the content of micron fibers, the higher the tensile strength.**

Figure 6. Tensile stress-strain curves of full-micron fiber and full-nanofiber BcF-CAs.

Fig. 1i Tensile stress-strain curve of the BcF-CAs.

Meanwhile, it can be noticed from the compressive stress-strain curves that the compressive strength of full-micron fiber aerogel (Figure 7), full-nanofiber aerogel (Figure 8), and BcF-CAs (Fig. 3a) are close to each other, but the energy dissipation rate of full-micron fiber aerogel is lower than that of full-nanofiber aerogel and BcF-CAs. In the Q15, we have demonstrated that the greater the energy dissipation, the smaller the stress wave propagation distance, which is not conducive to damage, and the material has lower brittleness and higher fracture strength. Therefore, it also indicated that the micron fiber component acts as a strong support while the nanofiber component functions as a significant stress dispersion. As shown Fig. 3a, incorporating appropriate micron fibers in the construction of BcF-CAs can significantly improve the tensile properties of the aerogels without compromising their compressive properties.

In addition, the content of micron fiber in BcF-CAs depends on the trade-off between tensile strength and bulk density. Given that the BcF-CAs are designed primarily for aerospace applications where ultra-lightweight materials are required,

choosing the appropriate micron fiber content is essential and desirable (answered in Q13).

Figure 7. Compressive stress-strain curves of full-micron fiber BcF-CAs.

Figure 8. Compressive stress-strain curves of full-nanofiber BcF-CAs.

Fig. 3a Compressive stress-strain curve of the BcF-CAs.

18. There are typos: a significant compression stain of 80% (page 14), Discussion heading should be Conclusion.

Response: We sincerely thank the reviewer for careful reading. We feel sorry for our carelessness.

We have corrected “Discussion” to “Conclusion” in our resubmitted manuscript. Thanks for your correction.

19. Which sample is mentioned in Fig. 3a-c?

Response: We sincerely thank the reviewer for the valuable comments, the comments are very helpful in improving our manuscript.

The sample mentioned in Fig. 3a-c is Supplementary Fig. 15.

Supplementary Figure 15. The anisotropic mechanical properties of the BcF-CAs.

20. Could the author double-check Young’s modulus in Fig. 3c? What is the strain used to determine the stress in that figure?

Response: We sincerely thank the reviewer for the valuable comments, the comments are very helpful in improving our manuscript.

(1) Could the author double-check Young’s modulus in Fig. 3c?

We apologize for the mistakes in the Manuscript. As suggested by the reviewer, we have double-checked and revised Young’s modulus in Fig. 3c and Supplementary Fig. 25 in the revised manuscript and Supporting Information. In addition, we also have carefully checked similar deficiencies throughout the manuscript.

Fig. 3 c Young's modulus, energy loss coefficient, and maximum stress as a function of the number of compression cycles.

Supplementary Figure 25. Young's modulus, energy loss coefficient, and maximum stress as a function of compression cycles

(2) What is the strain used to determine the stress in that figure?

We apologize for not describing Fig. 3c enough, which caused the reviewer to be confused.

The strain that determines the stress in Fig. 3c is from the cyclic test in Fig. 3b.
The data provided in Fig. 3c is a supplement to the compressive cycling tests in Fig. 3b.

21. Does changing the torsion angle affect the density and thermal conductivity of the material?

Response: We sincerely thank the reviewer for the valuable comments, the comments are very helpful in improving our manuscript.

(1) Theoretically, varying the twist angle does not affect the density of a material. The density is a function of the volume and weight of the material. However, changing the twist angle does not change the volume and weight, so the density of the material is not varied.

(2) As shown in Fig. 4b in the manuscript, the thermal conductivity increased with the density of the BcF-CAs. However, we have known that the torsion angle did not affect the density of the BcF-CAs. Consequently, the thermal conductivity is unchanged.

Fig. 4b Thermal conductivity of the BcF-CAs as a function of density.

22. Please provide reference and thermal conductivity (if possible) for Supplementary Table 1.

Response: We sincerely thank the reviewer for the valuable comments, the comments are very helpful in improving our manuscript.

Supplementary Table 1. The relevant properties of BcF-CAs and other insulation aerogels materials.

Materials	Maximum working temperature (°C)	Volume density (g/cm ³)	Tensile strength (MPa)	Thermal conductivity (Wm ⁻¹ K ⁻¹)
BcF-CAs	1200	0.211-0.283	98-213	0.037
SiO ₂ nanofibrous aerogels by freeze-shaping ¹⁰	1100	0.012-10	0.00417	0.023-0.032
ZrO ₂ -SiO ₂ nanofibrous aerogels by immerse-stacking ²⁶	1100	0.026	0.34	0.024

Binary-network structured SiO₂ nanofibrous aerogels (BSA)²⁷	1100	0.001	0.00214	0.2196- 0.28
Multifunctional core-shell nanorod aerogels²⁸	1200	0.128	1.6	0.024
SiC nanowire aerogel²⁹	900-1200	0.005-0.057	0.017-0.09	0.026
Hypocrystalline zircon nanofibrous aerogels (ZAGs)³⁰	1300	0.026	0.0424	0.026
Mullite nanofibrous aerogels via 3D reaction electrospinning (ICCA)¹³	1400	0.022-0.027	0.0127	0.0228
Porous carbon material³¹	350	0.011	0.00045	/
Carbon aerogels³²	800	0.0057	0.007	/

Comments from Reviewer #2:

In the present manuscript, the authors demonstrate the “Bouligand” (or twisted) structure that is seen in nature is suitable for strengthening fiber-based porous materials. The idea is interesting and rather easy to realize, and the tensile strength of the freeze-dried aerogel-like material is impressively high. In my opinion, however, in addition to the enhancement of mechanical strength, the authors have to report the thermal conductivity at high temperatures since the authors mention applications in the aerospace industry. To gain enough novelty to be accepted by Nature Communications, this reviewer requests to clarify this point (comment 5 below), together with some other revisions.

Response: We sincerely thank the reviewer for the positive and constructive comments.

1. The preparation process should be briefly written in the main text so that readers can understand what the materials are like. Especially for understanding how the angles were made between the layers and what is the AIBSi sol and its role.

Response: We sincerely thank the reviewer for the valuable comments, the comments are very helpful in improving our manuscript.

We apologize for not providing details about the fabrication process of BcF-CA, which makes the reviewers puzzled. In fact, we have added a schematic illustration of the manufacturing process in response to the Q5 of the first reviewer (Supplementary Fig. 4). Additionally, regarding the other concerns of the reviewer, we have provided detailed responses as follow:

(1) Preparation process and what is the AIBSi sol and its role.

Page 7, Line 148:

“In the proof-of-concept study, mullite nanofibers and Al₂O₃ macrofibers were carefully selected as the sample materials due to their superior thermal stability.¹ Fig. 1e and Supplementary Fig. S4 present the fabrication process of BcF-CAs. The fabrication strategy started with electrospinning mullite/poly (ethylene oxide) (PEO) sol to produce flexible mullite nanofibers (Supplementary Method). The obtained mullite nanofiber had a diameter of 310-420 nm (Supplementary Fig. S5) and a tensile strength of 0.47 MPa (5% strain) (Supplementary Fig. S6). The Al₂O₃ macrofibers (Supplementary Fig. S7) possessed a fiber diameter of ~7 μm and displayed an impressive tensile strength of 2.1 GPa. Then, the ceramic micro/nanofibers were subjected to immersion in AIBSi sol for 2 h. Notably, AIBSi is known for its remarkable thermal stability and mechanical properties, making it a “ceramic glue” for bonding

adjacent fibers. This cross-linking method relied on the formation of silicate bonds (X-O-Si), achieved through the calcination of silica nanofibers in the presence of oxygen.¹ Finally, the immersed macrofibers and nanofibers were arranged layer-by-layer with a specific helical angle (15°, 30°, 70°, 0°) and freeze-dried for ~48 h to obtain unbonded BcF-CAs. After annealing in a muffle furnace (900 °C for 1 h in flowing air), bonded BcF-CAs were obtained.”

(2) About how the angles were made between the layers:

We developed a “compass” (just like Fig.9b) guided layer-by-layer method for preparing spiral laminated samples with different interlayer twist angles. During the layer-by-layer lay-up process, each rectangular sheet was positioned at the diagonal incremental markers at the corners of two diameters opposite each other. Subsequently, the sheets were stacked layer-by-layer according to the variation in helix angle between each consecutive layer (Figure 9b). Apparently, parallel (0°/180°) laminates were easy to construct. However, helical stacked structure samples with other twist angles need more precise positioning during lay-up preparation. By implementing this preparation method, we ensure positioning inaccuracy remains below 1.5°.

However, since this method is only available in the laboratory, we intend to develop an automated lay-up machine with a motorized rotating table, offering the flexibility to adjust the twist angle based on sample requirements (Figure 9a).

Figure 9. (a) An automated lay-up machine with a motorized rotating table. (b) The compass that guided the layer-by-layer method in our manuscript.

2. Why were the two precursors of aluminum chloride and aluminum isopropoxide used?

Response: We sincerely thank the reviewer for the valuable comments, which are very helpful in improving our manuscript.

Initially, Bulent E. Yolda^{33,34} employed aluminum isopropoxide (AIP) and aluminum sec-butoxide as raw materials in conjunction with **nitric acid or acetic acid as peptizing agents to prepare γ -AlOOH sols**. These γ -AlOOH sols were subsequently utilized as precursors for producing transparent bulk alumina materials. The pioneering work played a pivotal role in advancing the sol-gel technique for the fabrication of thin films and bulk materials, marking a significant milestone in the field of sol-gel technology. Subsequently, Clerk et al.³⁵ prepared alumina sols based on the work of Yoldas, using **aluminum nitrate instead of nitric acid as the gelling solvent**. They found that **the gelling effect of aluminum nitrate on boehmite particles was more intense compared to nitric acid**, and the preparation time of boehmite sols could be significantly reduced by using aluminum nitrate as the gelling solvent.

Drawing inspiration from these previous studies, Nishio et al.³⁶ made innovative advancements by employing **aluminum nitrate, AIP, and tetraethoxysilane (TEOS)** as raw materials, with water as the solvent. By adopting this approach, they **successfully prepared aluminum-silica sols with a lower crystallization temperature for mullite**. Notably, this method offered the novel advantage of **achieving homogeneous mixing of alumina and silica components at the molecular level** without needing organic solvents.

Motivated by the above findings, we propose utilizing **AIP as the primary source of aluminum, TEOS as the silicon source, and alumina chloride as both the catalyst and peptizing agent** to achieve a molecularly homogeneous blending of alumina and silica components. The experimental results confirmed that the generated mullite sols were highly stable, with gelation times ranging from **5-7 days**. The following is the mechanism of **alumina chloride as the catalyst and peptizing agent**:

① The Mechanism of Catalysis

First, aluminum ion hexahydrate hydrolyzes in water (eq 1), releasing hydrogen ions in the solution and making the solution acidic, **which can catalyze the hydrolysis reaction of AIP and TEOS**. (eq2-3)

Scheme 2. The hydrolysis reaction of AIP and TEOS by the catalyzing of hydrogen ions.

The partially hydrolyzed metal-alcohol salts then undergo polycondensation reactions as shown in (eq 4-6,) and form silica-aluminum copolymers containing -Si-O-Si-, -Al-O-Al- and -Si-O-Al- structures.³⁷

R: OCH(CH₃)₂ or OCH₂CH₃

② The Mechanism of Peptizing

Additionally, inorganic metal salt ions have been identified as **peptizing** agents. Typically, TEOS and aluminum alkoxide have been widely utilized as the primary precursor materials for the sol-gel synthesis of mullite fibers. **Nevertheless**, it should be noted that the **hydrolysis rate of aluminum alkoxide surpasses that of TEOS by a substantial margin**, leading to an **undesirable chemical heterogeneity** within the gel fibers. This chemical non-uniformity adversely influences the high-temperature characteristics of the resulting mullite fibers, particularly in terms of creep resistance. **To address this issue**, the incorporation of inorganic metal salts, such as **aluminum chloride and aluminum nitrate**, has been explored as **potential peptizing agents**. Following the Derjaguin-Landau-Verwey-Overbeek theory, introducing **peptizing agents** promotes the formation of a **double-ionic layer between the alumina particles**. The aggregate stability of this system is facilitated by the presence of **“hydrophilic repulsion”**, ultimately leading to the generation of **homogeneous and stable sols**.

(Figure 10). **Consequently,** the utilization of such stabilizing agents facilitates the formation of highly **homogeneous and stable** sols.

Figure 10. Double electron layer structure model of aluminum-silica sol.

As shown in **Figure 10**, within the **double-ionic layer structure**, the gel nucleus consists of the aluminum-silicon copolymer, and there is Al^{3+} in the solution, so the gel nucleus will preferentially adsorb Al^{3+} , thus making the surface of the gel nucleus positively charged. Concurrently, a fraction of the negatively charged Cl^- ions are adsorbed in the vicinity of the gel nucleus, **forming an adsorption layer** that, in conjunction with the nucleus, constitutes a micelle. Notably, the number of charged ions adsorbed by the micelles is larger than the number of oppositely charged present ions in the adsorbed layer, thereby determining the charge polarity of the micelles based on the adsorbed ions. Surrounding the gel cluster, a limited number of Cl^- ions assemble to form the diffusion layer of sol. **The presence of a two-electron layer endows the aggregates with stability by the mechanism of “hydrophilic repulsion”, thereby facilitating the formation of homogeneous and stable sol.**

3. Please specify the role of the AIBSi sol. Fig. 1i-k describes a surface layer formed on the nanofibers, and silicate bonds (X-O-Si) form at elevated temperatures. However, I cannot believe the firm bonds form in between the 1-nm layers, so it is desired to show SEM images that prove the formation of the crosslinks. By the way, is there a peak of B in XPS? It seems so small that it is impossible to see in the survey spectrum.

Response: Thanks for the constructive comments, and we fully understand the reviewer's concern about forming firm bonds between the layers.

(1) So it is desired to show SEM images that prove the formation of the crosslinks

First, we sincerely apologize for any confusion that may have been caused to the reviewers by our inaccurate presentation. In fact, the AIBSi layer **on the surface of an unbonded single fiber** is only **1-2 nm thick**, **but at bonded sites, a 5-10 nm thick bonding layer can be formed (Fig. 1i)**. Such behavior can be attributed to **the tendency of AIBSi sol to accumulate at the fiber overlaps and agglomerate** during the

impregnation process, which subsequently leads to the formation of stable bonding points after freeze-drying and high-temperature calcination. More FE-SEM images were added in the revised Supporting Information (**Supplementary Fig. 12**)

It is noteworthy that an ultrathin bonding layer of only 1-2 nm formed on the surface of each fiber, limiting the possibility of over-bonding and excessive compactness of the aerogel.

Considering these comments above, we have updated Fig. 1h-i in the revised manuscript.

Fig. 1 Structural design and fabrication of BcF-CAs. (h) FE-SEM images showing the microscopic architectures of BcF-CAs. (i) A thin coating of AlBSi ceramic at bonded sites of the fibers.

Supplementary Figure 12. FE-SEM images of several representative bonding points between the nanofibers of BcF-CAs

(2) By the way, is there a peak of B in XPS? It seems so small that it is impossible to see in the survey spectrum.

Second, we apologize for the XPS spectrum of BcF-CAs, which led to the reviewer's uncertainty. In response to the reviewer's concerns, we have recharacterized the sample with XPS analysis in the revised manuscript (Fig. 1k). Notably, the relatively low concentration of elemental B (with a molar ratio of TEOS: AlCl₃·6H₂O: boric acid: H₂O = 1: 0.257: 0.059: 13.158) limits its visibility in the entire XPS spectrum. Hence, to address this issue, we have provided a fine spectrum of B in the revised Supplementary Information (Supplementary Fig. 13), which confirms the presence of B in the BcF-CAs.

Fig. 1 Structural design and fabrication of BcF-CAs. (k) XPS spectrum of BcF-CAs for all elements. a.u., arbitrary unit.

Supplementary Figure 13. XPS fine spectrum scan of B_{1s} BcF-CAs with the binding energy ranging from 182 to 196 eV. The black line represents the Background baseline of the fine spectrum scan: a.u., arbitrary unit.

4. On page 17, “For the BcF-CAs, λ_{conv} is severely restricted due to the blocking of air in single micropores,” but there is no discussion on the micropores. Instead, from SEM images, the pores are in the micrometer order so they should be called macropores.

***Response:** We thank the reviewer for the valuable comments, the comments are very helpful in improving our manuscript.*

We sincerely apologize for our carelessness. As pointed out by the reviewer, the pore size of the BcF-CAs indeed be classified as macropores (> 50 nm). We have appropriately revised the manuscript (Page 17, Line 375). Additionally, we have thoroughly double-checked the manuscript to avoid similar problems. Once again, we thank the reviewers for their careful suggestions.

Page 17, Line 375:

“For the BcF-CAs, λ_{conv} was severely restricted due to the blocking of air in individual macropores.”

5. The fatal drawback of this manuscript is there is no report on the thermal conductivity at elevated temperatures. The thermal conductivity values at room temperature are reported, but the novelty of this material is its mechanical strength at high temperatures like 1200 deg C, and the authors mention the use as an insulation layer in the aerospace industry (Fig. 4a). Therefore, it is needed to show the thermal conductivity values at such high temperatures, without which readers cannot judge this material is suitable for such applications or not. In addition, since the material loses strength above 1300 deg C, it must be used at 1200 deg C or lower. The authors should verify that the 1200 deg C is high enough for those applications or not. I am not sure how high the temperature is in the exhaust of the rocket engines. Also, the high-temperature thermal conductivity must be lower compared to other insulation materials, and the authors have to prove the value is low enough for such applications.

***Response:** We sincerely thank the reviewer for the valuable comments, the comments are very helpful in improving our manuscript.*

(1) The authors should verify that the 1200 deg C is high enough for those applications or not.

Before answering the reviewer's question, I would like to introduce the cooling technologies of the rocket engine and nozzle.

The engine, often referred to as the “heart” of the rocket, serves as the primary power source that propels the rocket into space. Through a series of cycles, fuel is burned in the main combustion chamber, resulting in high temperatures, volumetric expansion, and high-velocity jetting that ultimately generates thrust. Remarkably, inside the main combustion chamber, the gas temperature can reach staggering levels, as high as 3500 K (Figure 11), which exceeds the melting point of most materials and is approximately half the temperature of the scorching surface of the sun.

To ensure that the engine operates normally and efficiently for a relatively long time, the engine generally possesses a cooling system. Typically, it involves various cooling methods, including but not limited to:

Figure 11. The temperature of the combustion chamber has far exceeded the melting point of the inconel alloy, reaching more than half the temperature of the sun.

① Thickened Metal Walls for Heat Dissipation and Cooling

Thickening the combustion chamber and nozzle walls with metal (titanium or nickel-based alloys) allows the engine to absorb more heat and dissipate it immediately before it reaches melting point. However, this method requires significant material consumption, increased strength requirements due to the highly uneven distribution of material thicknesses, and, most fatally, redundant mass, which is a considerable challenge and potential detriment to the overall efficiency of the engine.

According to the **Tsiolkovsky rocket equation**:

$$\Delta V = V_e \ln\left(\frac{m_0}{m_1}\right)$$

Where ΔV is the increased velocity of the rocket, V_e is the emission velocity, i.e., the velocity at which the rocket fuel and oxidizer are ejected, and m_0 is the initial mass of the rocket, m_1 is the sum of the pure mass of the rocket after acceleration. **This equation shows that the dry mass ratio $\left(\frac{m_0}{m_1}\right)$ directly determines the velocity increment of the rocket. Therefore, to avoid excessive mass of the equipment, only the combustion chamber is considered using this type of heat dissipation.**

② Adjusting the Fuel Mixing Ratio

The temperatures at which fuel burns depend on the fuel-to-oxidizer ratio. The ratio that allows the fuel and oxidizer to react and burn fully is based on the “stoichiometric ratio” of the reaction equation. However, complete combustion results in high temperatures, which places greater demands on the cooling system of the engine. Therefore, to reduce the temperatures in the pre-combustion chamber and gas generator, similar mixing ratio variations are applied to the pre-combustion in the engine gas generator cycle, and the staged combustion cycle and the full-flow staged

combustion cycle are designed with an oxygen-rich pre-combustion chamber and a combustion-rich pre-combustion chamber. For example, the SSME RS-25 engine turbopump is designed with a mixing ratio deviation for rich combustion, and the Soviet NK-33 closed-cycle engine turbopump is designed with oxygen enrichment.

③ Ablative Cooling

Ablative cooling is a method of reducing nozzle temperature by vaporizing the outer material and dissipating the heat. Typically, an ablative cooling nozzle consists of a high-melting-point carbon composite material, which is ablated at high temperatures to manage thermal energy effectively. Uniquely, this cooling approach eliminates the requirement for additional engine components and strictly follows the physicochemical principles of self-regulation of the external environment temperature feedback, which makes the cooling effect better and more reliable.

④ Fuel/Cold Air Cooling

Fuel/cold air cooling is achieved by injecting a layer of liquid (cold fuel)/cold air between the combustion chamber wall and the nozzle wall to isolate the two high-temperature objects, thereby preventing excessive temperatures and thermal shock. To increase the contact area for better thermal transfer, the engine nozzle wall is filled with numerous tiny fuel/cold gas channels. The mainstream manufacturing method currently used is to cut out the corresponding tubes in the nozzle wall and seal them with metals such as copper and nickel alloys with high thermal conductivity to form fuel channels.

Figure 12. The diagram of fuel/cold air cooling and *the position of the insulating layer*.

The above cooling methods reduce the temperatures of the engine and nozzles, and they are generally about 1000 °C. In fact, reaching the ideal temperature will inevitably increase the weight of the machine and reduce the efficiency of fuel use, which is not conducive to the flight of the aircraft and is a waste of energy. Therefore,

an additional insulation layer is required. BcF-CAs have the advantages of high-temperature resistance and low density, which makes them an ideal insulation layer (Figure 12). The insulation layer refers to the thermal insulation between the high-temperature resistant (titanium, nickel-based alloy, fiber-reinforced ceramic-based composite) inside the nozzle and the non-high-temperature resistant outside, and it can be used as a backer for the thermal shock resistance. In other words, our material (BcF-CAs) can satisfy the use of high-temperature hot-end parts, which not only leads to a significant weight reduction but also saves cold air, improving the total pressure ratio, increasing the operating temperature of 400 - 500 °C based on the traditional cooling method, and reducing the weight of the structure by 50% - 70%, making it the critical thermal structure material for the upgrading of aircraft engines.

(2) It is needed to show the thermal conductivity values at such high temperatures, without which readers cannot judge this material is suitable for such applications or not. In addition, since the material loses strength above 1300 deg C, it must be used at 1200 deg C or lower. Also, the high-temperature thermal conductivity must be lower compared to other insulation materials, and the authors have to prove the value is low enough for such applications.

Given the above discussion, it is known that the temperature of space engines such as rockets is around 1000 °C after cooling through a series of cooling systems, in which case BcF-CAs can be applied. Then, as suggested by the reviewer, we tested the high-temperature thermal conductivity λ_t of BcF-CAs. As shown in Figure 13, the λ_t of the BcF-CAs was only $0.076 \text{ m W m}^{-1} \text{ K}^{-1}$ at 300°C. Meanwhile, λ_t increased with the higher temperature, reaching $0.225 \text{ m W m}^{-1} \text{ K}^{-1}$ at 1200 °C. This is due to the following two reasons:

- i. The current predominant method for high temperature thermal conductivity testing is the Hot Wire Method^{26,30,38-39}, but it is limited to a temperature range of less than 1000 °C. Therefore, we choose the Protective Hot Plate Method for testing. However, this method inevitably has some deviations from the Hot Wire Method, resulting in our data being larger than the actual values.
- ii. The λ_{rad} is usually negligible at temperatures below 100 °C, but becomes the main source for the thermal conductivity of BcF-CAs at high temperatures. λ_{rad} can be calculated by

$$\lambda_{rad} = \frac{\frac{16}{3} n^2 \sigma T_r^3}{e \cdot \rho} \quad (7)$$

where λ_{rad} is the conductivity of the thermal radiation, n is the refractive index of the heat transfer medium, σ is the Stefan-Boltzmann constant, e is the absorption coefficient (mass attenuation coefficient), and ρ is the density of the heat transfer medium. From Eq. (7), the λ_{rad} scaled linearly as $\lambda_{rad} \sim T_r^3$, represents a large λ_{rad} under high temperature conditions. These results indicate the exceptional thermal insulation capabilities of BcF-CAs even under extreme conditions and accentuate their potential for application as highly efficient thermal insulating materials.³⁸

Figure 13. Thermal conductivity of BcF-CAs as a function of temperature (steady-state thermal measurement in air).

Notably, despite the rather high λ_t measured on our BcF-CAs using **Protective Hot Plate Method**, it is comparable to the thermal conductivity of current thermal insulation materials (Fig. 14).^{26, 38} Therefore, in summary, our materials are fully suitable as potential candidates for aerospace insulation materials.

We tried our best to improve the manuscript and made some changes marked in red in the revised paper, which will not influence the content and framework of the paper. We appreciate the Editors/Reviewers' warm work earnestly and hope the correction will be approved. Once again, thank you very much for your comments and suggestions.

References

- 1 Chen, S. et al. Biomimetic discontinuous Bouligand structural design enables high-performance nanocomposites, **Matter** 5, 1563 (2022).
- 2 Yaraghi, N. et al. A sinusoidally architected helicoidal biocomposite, **Adv. Mater.** 28, 6835 (2016).
- 3 Wu, K. et al. Discontinuous fibrous Bouligand architecture enabling formidable fracture resistance with crack orientation insensitivity. **Proc. Natl. Acad. Sci.** 117, 15465-15472 (2020).

- 4 Nikolov, S. et al. Revealing the design principles of high-performance biological composites using ab initio and multiscale simulations: the example of lobster cuticle. *Adv. Mater.* 22, 519 (2010).
- 5 Guarín-Zapata, N. et al. Shear wave filtering in naturally-occurring Bouligand structures. *Acta Biomater.* 23, 11 (2015).
- 6 Weaver, J. C. et al. The stomatopod dactyl club: a formidable damage-tolerant biological hammer. *Science* 336, 1275-80 (2012).
- 7 Xiong, R., Wu, W., Lu, C., Cölfen, H. Bioinspired chiral template guided mineralization for biophotonic structural materials. *Adv. Mater.* 34, 2206509 (2022).
- 8 Yang, T. et al. High strength and damage tolerance in echinoderm stereom as a natural bicontinuous ceramic cellular solid. *Nat. Commun.* 13, 6083 (2022).
- 9 Gantenbein, S. et al. Three-dimensional printing of hierarchical liquid-crystal-polymer structures. *Nature* 561, 226-230 (2018).
- 10 Si, Y., Wang, X., Dou, L., Yu, J., Ding, B. Ultralight and fire-resistant ceramic nanofibrous aerogels with temperature-invariant superelasticity. *Sci. Adv.* 4, eaas8925 (2018).
- 11 Zhang, X. et al. Ultrastrong, superelastic, and lamellar multiarch structured ZrO₂-Al₂O₃ nanofibrous aerogels with high-temperature resistance over 1300 °C. *ACS Nano* 14, 15616-15625 (2020).
- 12 Pierre, A. C., Pajonk, G. M. Chemistry of aerogels and their applications. *Chem. Rev.* 102, 4243-4265 (2002).
- 13 Cheng, X., Liu, Y., Si, Y., Yu, J., B. Ding. Direct synthesis of highly stretchable ceramic nanofibrous aerogels via 3D reaction electrospinning. *Nat. Commun.* 13, 2637 (2022).
- 14 Xu, X. et al. Double-negative-index ceramic aerogels for thermal superinsulation. *Science* 363, 723-727 (2019).
- 15 Yang, L., Wang, J., Liu, H., Jiang, R., Cheng, H. Sol-gel temperature dependent ductile-to-brittle transition of aluminosilicate fiber reinforced silica matrix composite, *Composites Part B* 119, 79-89 (2017).
- 16 Jiang, R., Yang, L., Liu, H., Sun, X., Cheng, H. High-temperature mechanical properties of NextelTM 610 fiber reinforced silica composites, *Ceram. Int.* 44, 15356-15361 (2018).
- 17 Jiang, R. et al. A multiscale methodology studying the sintering temperature dependent mechanical properties of oxide matrix composites. *J. Am. Ceram. Soc.* 101, 3168-3180 (2018).
- 18 Johnson, D. D. Nextel 312 ceramic fiber from 3M. *J. Ind. Text.*, 11, 282-296 (1982).
- 19 Vaidya, R. U. et al. Effect of fiber coating on the mechanical properties of a Nextel 480 fiber reinforced glass matrix composite. *Mater. Sci. Eng. C*, 150, 161-169 (1992).

- 20 Petry, M. D., Mah, T. Effect of thermal exposures on the strengths of Nextel™ 550 and 720 filaments. *J. Am. Ceram. Soc.* 82,2081-2807 (1999).
- 21 Wilson, D. M., Visser, L. R. High performance oxide fibers for metal and ceramic composites. *Compos. Part A Appl. Sci. Manuf.* 32,1143-1153 (2001).
- 22 Hay, R. S., Fair, G. E., Tidball, T. Fiber strength after grain growth in Nextel™ 610 alumina fiber. *J. Am. Ceram. Soc.* 98, 1907-1914 (2015).
- 23 Meza, L. R., Das, S., Greer, J. R. Strong, lightweight, and recoverable three-dimensional ceramic nanolattices. *Science* 345, 1322-1326 (2014).
- 24 Weaver, J. C. et al. The stomarod dactyl club: a formidable damage-tolerant biological hammer. *Science* 336, 1275-80 (2012).
- 25 Suksangpanya, N., Yaraghi, N. A., Kisailus, D., Zavattieri, P. Twisting cracks in Bouligand structures. *J Mech Behav Biomed Mater* 76, 38-57(2017).
- 26 Zhang, X., Cheng, X., Si, Y., Yu, J., Ding, B. Elastic and highly fatigue resistant ZrO₂-SiO₂ nanofibrous aerogel with low energy dissipation for thermal insulation. *Chem. Eng. J.* 433, 133628 (2022).
- 27 Dou, L., X. Cheng, X., Zhang, Si, Y., Yu, J. & Ding, B. Temperature-invariant superelastic, fatigue resistant, and binary-network structured silica nanofibrous aerogels for thermal superinsulation. *J. Mater. Chem. A* 8, 7775-7783 (2020).
- 28 Liu, F. et al. Carbon layer encapsulation strategy for designing multifunctional core-shell nanorod aerogels as high-temperature thermal superinsulators. *Chem. Eng. J.* 455, 140502 (2023).
- 29 Su, L. et al. Ultralight, recoverable, and high temperature-resistant sic nanowire aerogel. *ACS Nano* 12, 3103-3111 (2018).
- 30 Guo, J. et al. Hypocrystalline ceramic aerogels for thermal insulation at extreme conditions. *Nature* 606, 909-916 (2022).
- 31 Gao, H. et al. A highly compressible and stretchable carbon spring for smart vibration and magnetism sensors. *Adv. Mater.* 33, 2102724 (2021).
- 32 Guo, F. et al. Highly stretchable carbon aerogels. *Nat. Commun.* 9, 881 (2018).
- 33 Yoldas, B. E. Hydrolysis of aluminium alkoxides and bayerite conversion. *J.appl. Chem. Biotechnol.* 23, 803-809 (1973).
- 34 Yoldas, B. E. Alumina gels that form porous transparent Al₂O₃. *J Mater Sci* 10, 1856-1860 (1975).
- 35 Clark, D. E., Dalzell, W. Jr., Adams, B. L. Inorganic salts as peptizing agents in the preparation of metal oxide sol-gel compositions. US 4801399 (1986).
- 36 Nishio T, Fujiki Y. Preparation of mullite fiber by a sol-gel method. *J. Ceram. Soc. Jpn.* 99, 654-659 (1991).
- 37 Pierre. A. C., Pajonk G. M. Chemistry of aerogels and their applications, *Chem. Rev.* 102, 11, 4243-4266 (2002).

- 38 Xu, X., Fu, S., Guo, J., Li, H., Huang, Y., Duan, X. Elastic ceramic aerogels for thermal superinsulation under extreme conditions, *Mater. Today* 42, 162-77 (2021).
- 39 Wang, F. et al. In situ synthesis of biomimetic silica nanofibrous aerogels with temperature-invariant superelasticity over one million compressions. *Angew. Chem. Int. Ed.* 59, 8285–8292 (2020).

REVIEWERS' COMMENTS

Reviewer #1 (Remarks to the Author):

The present manuscript is good to be published.

Reviewer #2 (Remarks to the Author):

The authors revised the manuscript in an effective way so now I recommend publication of this manuscript in Nat Commun.